# ON THE WINGS OF IMAGINATION: CONFLICTING SCRIPT-BASED MULTI-ROLE FRAMEWORK FOR HUMOR CAPTION GENERATION

**Wenbo Shang[1], Yuxi Sun[1], Jing Ma[1], Xin Huang[1,*]**
[1]Hong Kong Baptist University
{cswbshang, csyxsun, majing, xinhuang}@comp.hkbu.edu.hk

## ABSTRACT

Humor is a commonly used and intricate human language in daily life. Humor generation, especially in multi-modal scenarios, is a challenging task for large language models (LLMs), which is typically as funny caption generation for images, requiring visual understanding, humor reasoning, creative imagination, and so on. Existing LLM-based approaches rely on reasoning chains or self-improvement, which suffer from limited creativity and interpretability. To address these bottlenecks, we develop a novel LLM-based humor generation mechanism based on a fundamental humor theory, GTVH. To produce funny and script-opposite captions, we introduce a **h**umor-the**o**ry-driven **m**ulti-role LLM collaboration framework augm**e**nted with humor **r**etrieval (HOMER). The framework consists of three LLM-based roles: (1) *conflicting-script extractor* that grounds humor in key script oppositions, forming the basis of caption generation; (2) *retrieval-augmented hierarchical imaginator* that identifies key humor targets and expands the creative space of them through diverse associations structured as imagination trees; and (3) *caption generator* that produces funny and diverse captions conditioned on the obtained knowledge. Extensive experiments on two New Yorker Cartoon benchmarking datasets show that HOMER outperforms state-of-the-art baselines and powerful LLM reasoning strategies on multi-modal humor captioning. Our codes are available at https://github.com/Shang-hub/HOMER-Official-Implementation.

## 1 INTRODUCTION

Multi-modal humor generation has emerged to be important for exploring whether large language models (LLMs) can handle human-level linguistic and cognitive complexity (Wang et al., 2025; Attardo, 2024; Oring, 2016; Horvitz et al., 2024; Cocchieri et al., 2025; Baluja, 2025). Funny caption generation is a typical task of multi-modal humor generation, which aims to generate a funny caption for a given image. This involves combining *visual understanding* of cartoons with *humor understanding*, *creative imagination*, and *stylistic expression* (Zhang et al., 2024; Wang et al., 2025), which is technically challenging and even difficult for human beings.

However, current LLMs have been validated to have a weak inherent humor generation mechanism (Mirowski et al., 2024; Horvitz et al., 2024; Pawar et al., 2025; Gorenz & Schwarz, 2024; Cocchieri et al., 2025; Jentzsch & Kersting, 2023). To improve the humor generation ability of LLMs, a few existing methods typically rely on generic prompting (Zhang et al., 2024; Chen et al., 2024), multi-hop reasoning for self-improvement (Zhong et al., 2024), or task-specific tuning (Wang et al., 2025) to better steer model outputs towards funnier captions. Unfortunately, these methods, solely guided by the LLM-inherent humor mechanism, capture surface humor language rather than deep humor logical reasoning and creative humor imagining, leading to limited creativity and originality. For example, consider the cartoon in Figure 1(a), current LLMs (e.g., GPT-4o) and the state-of-the-art CLoT (Zhong et al., 2024) demonstrate a good ability to generate a semantic caption for describing the meeting, table, and caffeine, but lack enough humor for funny and deep imagination.

---

*Corresponding Author.

**(a) Funny Caption Generation**

**(b) Funny Caption Generation with GTVH Mechanism**

Figure 1: A comparison of our HOMER with GPT-4o and CLoT models in funny caption generation.

To address the above limitations, we propose a novel LLM-based humor generation framework, leveraging the well-established General Theory of Verbal Humor (GTVH) to generate script-opposite humor captions (Attardo & Raskin, 1991; Ruch et al., 1993; Attardo, 2016; Oring, 2016; Shang et al., 2022). The GTVH models humor creation through several interconnected knowledge resources, offering a natural fit for image-based humor caption generation. Therefore, our GTVH-based method centers on script opposition, enabling it to capture diverse humor mechanisms and generalize across a broad range of various images. Continue the above example of a cartoon in Figure 1, the *situation* is an office meeting, with the *script opposition* between ordinary coffee cups and oversized ones (see Steps 1-2), which establish the core logic foundation for humor generation. The *target* of humor is the oversized cups (see Step 3), which disrupt the expected norm. The imagination of the *target* operates through an associative chain (coffee → milk →cow), amplifying the absurdity (see Step 4). The *narrative strategy* frames this exaggeration as a visual twist, while the *language* condenses it into a funny caption (see Step 5) as shown in Figure 1(b). The humor point of Figure 1(b) lies here. In terms of references (e.g., person, tables, and chairs), the size of coffee cups is super large. Thus, the key conflicting script, i.e., gigantic coffee cups vs. normal ones. The ground-truth humorous caption is "Could you please pass me a cow?", highlighting that large coffee cups need a large amount of milk, which is even needed to produce by the whole cow. This is ridiculously abnormal and funny. Our generated caption, "HR says we can expense a cow now", which has a similar humor effect as the ground-truth and even playfully exaggerating workplace coffee consumption by involving the expense department of Human Resources (HR). As a result, our caption has a better humorous effect than that of GTP-4o and CLoT in Figure 1(a).

Technically, we propose a **h**umor-the**o**ry-driven **m**ulti-role LLM collaboration framework augm**e**nted with humor **r**etrieval (HOMER) for funny image caption generation. HOMER identifies a clear and interpretable humor mechanism reliant on the collaboration of three roles of LLMs: **Conflicting-script Extractor** extracts a detailed situation description from the image and analyzes contrast and incongruity elements based on the definition of script opposition, capturing the core humor logic and essential humor creativity. The result of the extractor is the basis of the generation process. **Hierarchical Imaginator** aims to identify and enhance the critical humor target in the image. To expand its creative space, the imaginator conducts the humorous imagination of targets through our designed imagination trees, which are built by multi-view associations with LLM and humor-relevance retrieval from our collected joke database. **Caption Generator** combines the detailed situation description, conflicting scripts, and diverse imaginative trees of targets to generate funny captions in a configuration of the five knowledge resources. Extensive experiments on two public New Yorker Cartoon benchmarks demonstrate the superiority of HOMER against state-of-the-art competitors by achieving ~7% improvement on average.

## 2 HOMER

In this section, we present the problem of funny caption generation and our framework, HOMER.

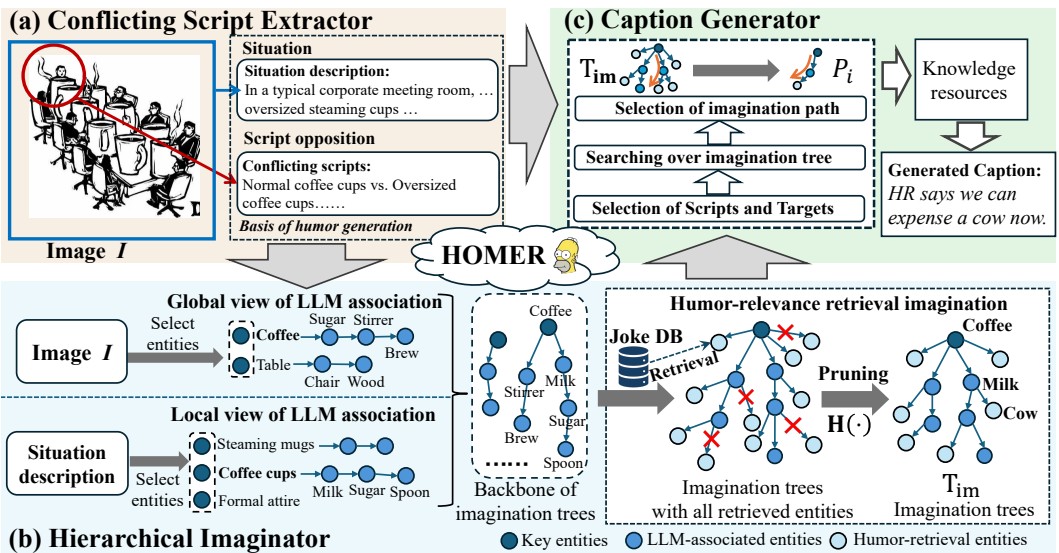

Figure 2: Framework of HOMER with three LLM-based roles: (a) Conflicting script extractor, deriving a detailed situation description and conflicting scripts as the basis of humor generation. (b) Hierarchical imaginator, identifying and enhancing the humor target with multi-view LLM associations and humor-relevance retrieval imagination. (c) Caption generator, generating funny and diverse captions conditioned on the obtained knowledge.

**Fundamental Humor Theory.** The GTVH humor theory is the theoretical foundation for our HOMER, modeling humor as the interaction of several knowledge resources: script opposition, situation, target, narrative strategy, and language (Attardo & Raskin, 1991). Central to humor is script opposition, which captures conflicts between semantic frames (scripts). It underlies humor by establishing expectations and then violating them, thereby enabling exaggeration or absurdity. In Figure 1 (b), the conflict between a professional office setting with unexpected gigantic cups juxtaposes scripts of routine and hyperbole, yielding a humorous reading. This script opposition leverages surprise, incongruity, and cognitive resolution, which are central to effective and engaging humor. More examples can be found in Section 3.4 and Appendix B.10.

**Problem Formulation.** Given an input image $I$, the funny caption generation task aims to generate a relevant and funny caption $\text{Cap}(I)$ for image $I$. The ground-truth of this task is composed of human-written funny captions. The goal of tackling this task is to assess whether the generated $\text{Cap}(I)$ derived by the multi-role models wins against human-written captions.

**Overview of HOMER.** The key idea of generating a humorous caption by our HOMER framework is to extract conflicting scripts from the given image and imagine script-opposition funny based on LLM association and joke database. As shown in Figure 2, HOMER contains three LLM-based roles, which are the conflict script extractor $\text{Extract}(\cdot)$, the hierarchical imaginator $\text{Imagine}(\cdot)$, and the caption generator $\text{Gen}(\cdot)$. $\text{Extract}(I) \rightarrow (\mathcal{C}, D)$ yields script oppositions $\mathcal{C}$ and a situation description $D$. $\text{Imagine}(I, \mathcal{C}, D) \rightarrow \mathcal{T}_{\text{im}}$ identifies key humor targets and derives target imagination tree. $\Omega \in NS \times LA$ sets the narrative strategy and selects linguistic style. With prompt $\Phi(\mathcal{C}, D, \mathcal{T}_{\text{im}}, \Omega)$, the generator generates $\text{Cap}(I) = \text{Gen}(\Phi(\mathcal{C}, D, \mathcal{T}_{\text{im}}, \Omega))$.

## 2.1 CONFLICTING SCRIPT EXTRACTOR

To ensure the extraction of precise and comprehensive conflicting scripts, our LLM-based conflicting script extractor $\text{Extract}(\cdot)$ first analyzes the scene in the given image $I$ and derives a script-opposition-central situation description $D$ as the contextual background of the funny caption (e.g., meeting room, professional figures, steaming cups, natural or serious expressions in Figure 2), including location, characters, facial expressions, and actions, while emphasizing inherent conflicting elements (e.g., oversized steaming cups) in $I$. Next, $\text{Extract}(\cdot)$ is designed to systematically identify and analyze conflicting or incongruous elements in the image $I$. As GTVH posits, the definition of script opposition is *the relation between two conflicting or contrasting semantic frames (scripts) in*

*a joke*. Building on this definition, we design a prompt $\Phi_{\text{script}}(\cdot)$ to guide $\text{Extract}(\cdot)$ to analyze and extract all relevant conflicting scripts that exist in the situation description $D$ and the image $I$. Formally, the set of conflicting scripts $\mathcal{C}$ is derived as $D = \text{Extract}(I)$, $\mathcal{C} = \text{Extract}(\Phi_{\text{script}}(I, D))$. $D$ and $\mathcal{C}$ serve as the foundation of the whole generation process.

## 2.2 HIERARCHICAL IMAGINATOR

Based on the constructed humor foundation, our hierarchical imaginator $\text{Imagine}(\cdot)$ first identifies a set of key entities $\{t_i\}$ described in $\mathcal{C}$ and $D$ as candidate humor targets of the funny caption. Then, to enrich knowledge about the identified targets and expand its creativity, $\text{Imagine}(\cdot)$ enhances each target $t_i$ by conducting diverse imaginative associations. To capture diverse and high-quality imaginative associations, $\text{Imagine}(\cdot)$ is designed as a hierarchical architecture to combine multi-view LLM free-associations with humor-relevance retrieval imagination to construct a set of imagination trees $\mathcal{T}_{\text{rm}}$. Particularly, multi-view LLM associations serve as deep-pattern imagination, forming backbone chains of $\mathcal{T}_{\text{rm}}$, while humor-relevance retrieval serves as broad-pattern imagination to expand $\mathcal{T}_{\text{im}}$ by discovering relevant humor associations in our collected joke database. When constructing $\mathcal{T}_{\text{im}}$, a humor-relevance score $\mathbf{H}(e_\tau^{(i)}, \varepsilon)$ is introduced to quantitatively measure the degree of humorous relevance between backbone entities $e_\tau^{(i)}$ and retrieved entities $\varepsilon$, pruning $\mathcal{T}_{\text{im}}$ and removing retrieved entities $\varepsilon$ with weak humor relevance.

**Identify candidate targets from local and global views.** Define the set of views $V = \{\text{loc,glob}\}$. The local observation $O_{\text{loc}}$ is from the detailed situation description $D$, capturing fine-grained entities or unexpected features within the image (e.g., oversized cups, professional figures). The global observation $O_{\text{glob}}$ leverages the image $I$ to emphasize the obvious entities in the scene (e.g., cups, table). Coarse-grained and fine-grained entities can evoke different LLM associations (e.g., coffee cups, tea, figured people, etc). For each view $v \in V$, the imaginator extracts $m$ entities from $O_v$ as candidate targets that are most relevant to conflicting scripts $\mathcal{C}$. Formally,

$$O_v \times \mathcal{C} \rightarrow \text{Ent}(O_v, C), \quad \text{Ent}(O_v, \mathcal{C}) = \{t_1, ..., t_m\}, \quad T_{\text{root}} = \{\text{Ent}(O_v, \mathcal{C}) | v \in V\}.$$

$m$ is dependent on LLM analysis of $O_v$ and $\mathcal{C}$. Identified candidate targets in $T_{\text{root}}$ serve as ancestor nodes of a forest of imagination trees, thereby guiding subsequent imaginative exploration.

**Deep imagination forms the backbone chains of $\mathcal{T}_{\text{im}}$.** Deep-pattern imaginative chains from each $t_i \in T_{\text{root}}$ are modeled as a first-order association process through an LLM-driven association function $f_{\text{chain}}(\cdot)$ with possible relations (e.g., ingredient, container, source, etc.) For an ordered free-association chain $T'_i = \langle e_0^{(i)}, e_1^{(i)}, ..., e_n^{(i)} \rangle$, the construction process is

$$e_{\tau+1}^{(i)} = f_{\text{chain}}(e_\tau^{(i)}), \quad \tau = 0, ..., n-1,$$

where $e_0^{(i)} = t_i$ and each successor $e_{\tau+1}^{(i)}$ is imagined solely from its direct predecessor $e_\tau^{(i)}$. The length $\tau$ is adaptively determined by LLMs with empirical average length $\mathbb{E}[\tau] \approx 4$. The recursive procedure $f_{\text{chain}}(\cdot)$ enables progressively deeper levels of imaginative reasoning, ensuring each entity is conditionally dependent on its predecessor. After constructing two views of backbone chains $\{T'_i | t_i \in \text{Ent}(O_v, \mathcal{C}), v \in V\}$, the imaginator merges local and global-view chains by aligning identical entities and removing duplicates. For example, entities "coffee" and "coffee cups" can be merged into "coffee cups". As a result, each candidate target $t_i$ is associated with a unique and multi-view imagination tree $T_i$, forming the backbone chains of imaginative trees $\mathcal{T}$.

**Broad imagination expands the imaginative chains of $\mathcal{T}_{\text{im}}$.** To expand the backbone of the imagination tree with relevant humor associations in daily life, we design a humor-relevance retrieval from our collected joke database $\mathcal{J}$, which is reorganized from 12 open-source joke datasets. First, the imaginator conducts *top-K relevant joke retrieval*. For each LLM-associated entity $e_\tau^{(i)} \in T_i$, the imaginator constructs a query embedding $\mathbf{z}_q = f_{\text{emb}}(D, \mathcal{C}, e_\tau^{(i)})$. For each joke $j \in \mathcal{J}$ with embedding $\mathbf{z}_j$, we calculate the cosine similarity $\text{sim}(\mathbf{z}_q, \mathbf{z}_j)$. The top-$k$ jokes are retrieved by ranking all $j$ according to $\text{sim}(\mathbf{z}_q, \mathbf{z}_j)$, i.e., $J_{\text{topK}} = \{j \in \mathcal{J} | sim(\mathbf{z}_j, \mathbf{z}_q) \geq sim(\mathbf{z}_{j'}, \mathbf{z}_q), \forall j' \in \mathcal{J} \setminus J_{\text{topK}}\}$, ensuring selected jokes are relevant to both the query entity $e_\tau^{(i)}$ and the foundation of humor $D$ and $\mathcal{C}$. $f_{\text{emb}}(\cdot)$ can be the statistical embedding method for efficiency, or other LM-based methods. Then, for each retrieved joke $j \in J_{\text{topK}}$, the imaginator tokenizes and lemmatizes $j$ into a set of tokens $\mathcal{E}_j$ as leaf nodes for the query node $e_\tau^{(i)}$. Finally, the imaginator conducts ***HOMER-pruning***

with a designed humor-relevance score $\mathbf{H}(e_\tau^{(i)}, \varepsilon)$ to filter out leaf nodes with weak humor relevance, deriving high-quality imagination trees $\mathcal{T}_{\mathrm{im}}$.

***HOMER-pruning.*** To filter out leaf nodes with weak humor relevance to $e_\tau^{(i)}$, we design a humor-relevance score $\mathbf{H}(e_\tau^{(i)}, \varepsilon)$, where $\varepsilon \in \mathcal{E}_j$. $\mathbf{H}(e_\tau^{(i)}, \varepsilon)$ builds on three key scores, which are relevance-opposition $\mathbf{H}_{\mathrm{rel}}(e_\tau^{(i)}, \varepsilon)$, humor-frequency $\mathbf{H}_{\mathrm{freq}}(\varepsilon)$, and POS-diversity scores $\mathbf{H}_{\mathrm{div}}(\varepsilon)$ as follows.

$$\mathbf{H}(e_\tau^{(i)}, \varepsilon) = \mathbf{H}_{\mathrm{rel}}(e_\tau^{(i)}, \varepsilon) + \mathbf{H}_{\mathrm{freq}}(\varepsilon) + \mathbf{H}_{\mathrm{div}}(\varepsilon). \tag{1}$$

We then retain tokens for which $\mathrm{rank}(\mathcal{H}(e_\tau^{(i)}, \varepsilon)) \leq \delta$, thereby pruning the imagination tree $T_i$, where $\delta$ is the desired rank threshold. $\mathrm{rank}(\mathcal{H}(e_\tau^{(i)}, \varepsilon))$ denote the rank of $\varepsilon$ according to its humor-relevance score for $\varepsilon \in \mathcal{E}_j$.

**Term-1: Relevance-Opposition score.** Inspired by GTVH, we design the relevance-opposition score $\mathbf{H}_{\mathrm{rel}}(e_\tau^{(i)}, \varepsilon)$ between entities $e_\tau^{(i)}$ and $\varepsilon$ as a joint function of semantic similarity and conceptual opposition, thereby capturing semantic relevance and surprise incongruity essential to humor. To accurately measure $\mathbf{H}_{\mathrm{rel}}(e_\tau^{(i)}, \varepsilon)$, we utilize WordNet (Miller, 1995), which affords structured semantic relations for reliable sense discrimination and similarity assessment. Specifically, target semantic similarity (TSS) is quantified using the Wu-Palmer similarity $\mathrm{Sim}_{\mathrm{wup}}(\cdot, \cdot)$. Let $S_{e_\tau}$ and $S_\varepsilon$ represent the sets of synsets associated with $e_\tau^{(i)}$ and $\varepsilon$. For $S_{e_\tau}, S_\varepsilon \neq \emptyset$,

$$\mathrm{TSS}(s_{e_\tau}, s_\varepsilon) = \max_{s_{e_\tau} \in S_{e_\tau}, s_\varepsilon \in S_\varepsilon} \mathrm{Sim}_{\mathrm{wup}}(s_{e_\tau}, s_\varepsilon). \tag{2}$$

Otherwise, $\mathrm{Sim}_{\mathrm{wup}}(s_{e_\tau}, s_\varepsilon) = 0$. Conceptual opposition (CO) is measured as the Jaccard dissimilarity between concept sets of $e_\tau^{(i)}$ and $\varepsilon$. For a given synset $s$, its concept set $\mathcal{R}(s)$ is defined as the union of its neighboring concepts, including its synonyms, hypernyms, hyponyms, meronyms, and holonyms, denoted by $\mathrm{Hyper}(s)$, $\mathrm{Hypo}(s)$, $\mathrm{Mero}(s)$, and $\mathrm{Holo}(s)$, respectively. Thus, $\mathcal{R}(s) = s \cup \mathrm{Hyper}(s) \cup \mathrm{Hypo}(s) \cup \mathrm{Mero}(s) \cup \mathrm{Holo}(s)$. The Jaccard overlap between senses $s_{e_\tau} \in S_{e_\tau}$ and $s_\varepsilon \in S_\varepsilon$ is

$$Jacco(s_{e_\tau}, s_\varepsilon) = \frac{|R(s_{e_\tau}) \cap R(s_\varepsilon)|}{|R(s_{e_\tau}) \cup R(s_\varepsilon)|}, \quad \text{if } |R(s_{e_\tau}) \cup R(s_\varepsilon)| > 0.$$

Otherwise, $Jacco(s_{e_\tau}, s_\varepsilon) = 0$. Therefore, we formulate the conceptual opposition as

$$\mathrm{CO}(s_{e_\tau}, s_\varepsilon) = 1 - \max_{s_{e_\tau} \in S_{e_\tau}, s_\varepsilon \in S_\varepsilon} Jacco(s_{e_\tau}, s_\varepsilon), \quad \mathrm{CO}(s_{e_\tau}, s_\varepsilon) \in [0, 1]. \tag{3}$$

$\mathrm{CO}(s_{e_\tau}, s_\varepsilon)$ treats opposition as low overlap between the lexical neighborhoods induced by core semantic relations. Thus, we model two opposing tendencies: semantic similarity $\mathrm{TSS}(s_{e_\tau}, s_\varepsilon)$, and conceptual opposition $\mathrm{CO}(s_{e_\tau}, s_\varepsilon)$. $\mathbf{H}_{\mathrm{rel}}(e_\tau^{(i)}, \varepsilon)$ should be designed based on three criteria: (i) dominated by semantic similarity; (ii) incorporating a similarity-gated, bounded bonus for conceptual opposition; and (iii) established through a principled balance between two opposing tendencies. To achieve it, we introduce a shaping function $f : [0, 1] \to [0, c]$ that is increasing and bounded, with $f(0) = 0$ and a moderate peak as similarity grows. Thus, based on Eq 2 and Eq. 3, the calculation of the relevance-opposition score can be formulated as

$$\mathbf{H}_{\mathrm{rel}}(e_\tau^{(i)}, \varepsilon) = \mathrm{TSS}(s_{e_\tau}, s_\varepsilon) + f(\mathrm{TSS}(s_{e_\tau}, s_\varepsilon))\mathrm{CO}(s_{e_\tau}, s_\varepsilon), \quad \text{with } f(x) = x \exp(-x). \tag{4}$$

Detailed proof of convergence and monotonicity can be found in Appendix C.

**Term-2: Humor-Frequency score.** $\mathbf{H}_{\mathrm{freq}}(\varepsilon)$ quantifies the importance of $\varepsilon$ based on its empirical occurrence frequency, defined as the geometric mean of token frequency and normalized joke frequency over $J_{\mathrm{topK}}$:

$$\mathbf{H}_{\mathrm{freq}}(\varepsilon) = \sqrt{\frac{\sum_{j \in J_{\mathrm{topK}}} \mathrm{count}(\varepsilon, \mathcal{E}_j)}{\sum_{j \in J_{\mathrm{topK}}} |\mathcal{E}_j|} \frac{\sum_{j \in J_{\mathrm{topK}}} \mathbb{I}[\varepsilon \in \mathcal{E}_j]}{|J_{\mathrm{topK}}|}} \tag{5}$$

$\mathrm{count}(\varepsilon, \mathcal{E}_j)$ is the multiplicity of $\varepsilon$ in $j$. The indicator function $\mathbb{I}(\cdot)$ equals 1 if the argument is true and 0 otherwise. Higher $\mathbf{H}_{\mathrm{freq}}(\varepsilon)$ indicates that $\varepsilon$ appears frequently and across many jokes in $J_{\mathrm{topK}}$, evidencing a statistically meaningful association with $e_\tau^{(i)}$ and $\mathcal{C}$ in context $D$.

**Term-3: POS-Diversity score.** $\mathbf{H}_{\mathrm{div}}(\varepsilon)$ assesses the lexical richness of $\varepsilon$ based on parts of speech (POS). $N_P$ denotes the POS inventory in WordNet. $N(\varepsilon)$ is the occurrence number of $\varepsilon$ tagged with $p \in P$. Higher $\mathbf{H}_{\mathrm{div}}(\varepsilon)$ indicates that token $\varepsilon$ has more lexical ambiguity, thereby creating more opportunities for puns and wordplay in funny captions. The normalized POS-diversity score is

$$\mathbf{H}_{\mathrm{div}}(\varepsilon) = \frac{\sum_{p \in P} \mathbb{I}[N(\varepsilon) > 0]}{|P|}. \tag{6}$$

Therefore, based on Eq. 4, Eq. 5, Eq. 6, and Eq. 1, $\mathcal{H}(e_\tau^{(i)}, \varepsilon)$ is calculated to retain leaf nodes for which $\mathrm{rank}(\mathcal{H}(e_\tau^{(i)}, \varepsilon)) \le \delta$, thereby deriving diverse and high-quality imagination trees $\mathcal{T}_{\mathrm{im}}$.

## 2.3 Caption Generator

Our caption generator $\mathrm{Gen}(\cdot)$ aims to generate funny and diverse captions $\mathrm{Cap}(I)$ for the given image $I$ based on the obtained knowledge. Specifically, $\mathrm{Gen}(\cdot)$ begins by randomly selecting key conflicting scripts $C \in \mathcal{C}$ and relevant humor target $t_i \in T_{\mathrm{root}}$ from all candidate targets. Note that not all candidate targets are used in the final caption. For each candidate target $t_i$, the associated imagination tree $T_i \in \mathcal{T}_{\mathrm{im}}$ is traversed to enumerate all possible paths from the ancestor node to the leaf nodes through depth-first search (DFS), denoted by $\mathcal{P}_i$. A single imagination path $P_i \in \mathcal{P}_i$ is then sampled, representing a creative chain of humorous associations. The generation prompt is constructed by integrating the situation description $D$, the selected conflicting scripts $C$, the creative imagination path $P_i$ of selected humor target $t_i$, and the generation options $\Omega \in NS \times LA$, where $\Omega$ specifies the narrative strategy and linguistic style. Formally, the prompt can be represented as $\Phi(\mathcal{C}, D, P_i, \Omega)$, which is then fed into the LLM-based caption generator producing the final funny caption, i.e., $\mathrm{Cap}(I) = \mathrm{Gen}(\Phi(\mathcal{C}, D, \mathcal{T}_{\mathrm{im}}, \Omega))$.

## 3 Experiments

**Datasets.** We evaluate the performance on two real-world New Yorker datasets, *Human in AI* (Zhang et al., 2024) and *Electronic sheep* (Hessel et al., 2023), including cartoon images, standard cartoon descriptions, humorous captions, and ranking of captions based on their humorous degree, as detailedly shown in Table 1. Following the settings in *Human in AI*,

Table 1: Statistics of datasets.

| Datasets | Human in AI | Electronic sheep |
|---|---|---|
| #Cartoons | 365 | 679 |
| Avg #captions | 6,044 | 6 |
| #Groups | 3 | 2 |
| Ranking | Global | Pairwise |
| Description | GPT-4o | Human |

we evaluate all models by comparing the generated captions against three groups of human-written captions at different ranking levels, which include #top10, #200-#209, #1000-#1009. As *Electronic sheep* has three ranking pairs of captions per cartoon, we split it into two groups. Higher ranking captions in all pairs form the High-Humor group. Otherwise, the Low-Humor group. In particular, we collect and reorganize 11 one-liner joke datasets through a multi-stage data processing as our humor retrieval dataset. We provide more details of our humor retrieval dataset in Appendix B.1.

**Competitors and Metrics.** We evaluate HOMER against four state-of-the-art models for humor generation: HumorousAI (Zhang et al., 2024), LoL (Wang et al., 2025), Phunny (Chen et al., 2024), and CLoT (Zhong et al., 2024). Additionally, we also compare with three widely-adapted and advanced reasoning strategies: chain of thought (CoT) (Wei et al., 2022), few-shot reasoning (Alayrac et al., 2022), and self-consistency (Wang et al., 2023). To assess the reliable measure for creative caption generation (Zhang et al., 2024; Hessel et al., 2023), we use the unbiased *Pass@K* metric to measure the probability that HOMER-generated humorous captions win the human-written caption over multiple $k$ trials (Liu et al., 2024; Mohammadi et al., 2025; Zhang et al., 2025; Yu et al., 2024).

$$\mathrm{pass@k} = \frac{1}{N} \sum_{i=1}^{N} \left[ 1 - \frac{\binom{n_i - c_i}{k}}{\binom{n_i}{k}} \right], \tag{7}$$

where $N$ denotes the total number of images, $n_i$ is the number of captions generated for the $i$-th image, and $c_i$ is the number of captions evaluated as funnier than the human caption. For comprehensive evaluation, we report the results at $K = \{1, 3, 5\}$. Each pass@K calculation is the average value of five trials. More details of competitors and metrics in Appendix B.2 and B.3.

Table 2: Performance (%) of funny caption generation (mean pass@k over 5 runs) on two datasets with four base LLMs (GPT-4o, Claude-4, Qwen-VL, and LLaVA-1.5). Higher scores are better.

| | Humor in AI | | | | | | | | | Electric sheep | | | | | |
| | #Top10 | | | #200-209 | | | #1000-1009 | | | High-Humor | | | Low-Humor | | |
| Method | @1 | @3 | @5 | @1 | @3 | @5 | @1 | @3 | @5 | @1 | @3 | @5 | @1 | @3 | @5 |
|---|---|---|---|---|---|---|---|---|---|---|---|---|---|---|---|
| **GPT-4o** | | | | | | | | | | | | | | | |
| CoT | 45.79 | 70.59 | 79.61 | 57.28 | 82.85 | 85.56 | 61.58 | 86.90 | 88.65 | 57.52 | 76.13 | 81.19 | 63.64 | 77.76 | 84.01 |
| Few-shot | 58.07 | 78.91 | 82.44 | 65.12 | 81.14 | 84.27 | 65.59 | 88.39 | 90.83 | 50.67 | 69.33 | 80.67 | 55.67 | 72.66 | 83.66 |
| Self-consistency | 62.03 | 77.96 | 82.93 | 68.09 | 84.45 | 87.72 | 69.42 | 85.51 | 88.93 | 48.57 | 64.95 | 74.23 | 62.02 | 70.47 | 78.78 |
| HumorousAI | 62.11 | 81.24 | 85.15 | 69.38 | 85.32 | 87.86 | 73.46 | 85.42 | 88.40 | 67.39 | 80.57 | 83.38 | 69.41 | 80.65 | 85.33 |
| LoL | 56.30 | 75.21 | 81.01 | 64.50 | 80.85 | 85.21 | 67.29 | 83.83 | 88.73 | 61.26 | 79.22 | 84.55 | 64.73 | 80.60 | 84.73 |
| Phunny | 16.09 | 27.47 | 32.94 | 20.38 | 34.23 | 41.25 | 23.80 | 38.74 | 45.99 | 26.22 | 36.13 | 45.92 | 29.31 | 38.32 | 48.05 |
| CLoT | 61.17 | 75.29 | 80.00 | 59.52 | 72.47 | 76.47 | 68.70 | 78.00 | 81.17 | 63.33 | 71.83 | 77.33 | 67.49 | 81.16 | 87.83 |
| Ours | **66.41** | **83.70** | **89.18** | **73.40** | **88.38** | **92.57** | **76.32** | **90.50** | **94.19** | **75.53** | **89.21** | **92.10** | **79.45** | **91.48** | **93.81** |
| Improv.(%) | +6.92 | +3.03 | +4.77 | +5.79 | +3.59 | +5.36 | +3.89 | +2.39 | +3.70 | +12.1 | +10.7 | +8.93 | +14.4 | +12.7 | +6.81 |
| **Claude-4** | | | | | | | | | | | | | | | |
| CoT | 37.31 | 47.62 | 51.01 | 40.03 | 48.70 | 51.01 | 42.27 | 41.87 | 51.67 | 57.52 | 69.00 | 72.51 | 63.50 | 74.39 | 78.01 |
| Few-shot | 61.67 | 78.70 | 82.67 | 70.00 | 85.13 | 88.67 | 69.19 | 83.70 | 87.00 | 32.67 | 54.00 | 66.67 | 48.33 | 63.67 | 68.33 |
| Self-consistency | 60.73 | 76.90 | 81.66 | 68.26 | 81.00 | 85.33 | 68.73 | 82.50 | 86.33 | 57.72 | 74.39 | 79.13 | 67.41 | 82.15 | 87.66 |
| HumorousAI | 62.86 | 78.86 | 82.67 | 70.39 | 83.46 | 86.33 | 68.66 | 82.06 | 85.98 | 59.40 | 77.66 | 83.33 | 65.69 | 83.13 | 89.83 |
| LoL | 58.06 | 75.19 | 80.00 | 68.40 | 84.40 | 88.67 | 67.06 | 83.30 | 87.89 | 60.60 | 79.33 | 83.00 | 66.23 | 83.04 | 87.66 |
| Phunny | 14.24 | 32.87 | 46.40 | 16.99 | 34.51 | 45.75 | 18.03 | 40.06 | 53.59 | 20.16 | 38.58 | 48.33 | 30.33 | 50.20 | 60.41 |
| CLoT | 43.15 | 53.25 | 56.00 | 50.91 | 59.05 | 62.00 | 53.65 | 61.44 | 63.00 | 41.67 | 62.00 | 68.33 | 51.83 | 72.83 | 79.16 |
| Ours | **64.67** | **82.67** | **87.00** | **71.27** | **86.33** | **90.33** | **71.06** | **85.47** | **89.00** | **62.27** | **81.37** | **86.94** | **71.75** | **89.86** | **95.19** |
| Improv. | +2.88 | +4.83 | +5.24 | +1.25 | +1.41 | +1.87 | +2.70 | +2.56 | +1.26 | +2.75 | +2.57 | +4.33 | +6.44 | +8.09 | +5.96 |
| **Qwen-VL (7B)** | | | | | | | | | | | | | | | |
| CoT | 16.76 | 27.33 | 33.01 | 25.46 | 38.69 | 44.44 | 22.85 | 35.11 | 40.63 | 19.06 | 30.90 | 36.66 | 26.83 | 37.39 | 41.83 |
| Few-shot | 19.60 | 29.59 | 33.67 | 27.67 | 39.57 | 44.33 | 26.46 | 38.79 | 44.67 | 19.38 | 30.96 | 35.73 | 28.69 | 40.44 | 45.19 |
| Self-consistency | 15.86 | 24.99 | 28.67 | 26.13 | 37.23 | 41.67 | 25.06 | 36.53 | 41.33 | 15.26 | 21.61 | 24.74 | 19.93 | 28.28 | 33.16 |
| HumorousAI | 18.26 | 27.53 | 30.67 | 27.67 | 38.90 | 43.33 | 25.40 | 37.00 | 41.33 | 11.80 | 16.83 | 18.99 | 16.56 | 25.56 | 29.16 |
| LoL | 15.12 | 23.84 | 28.17 | 19.86 | 32.61 | 38.14 | 21.37 | 34.29 | 40.54 | 17.58 | 28.53 | 31.50 | 24.94 | 38.24 | 44.13 |
| Phunny | 5.92 | 9.25 | 11.11 | 6.18 | 10.37 | 14.81 | 2.96 | 8.89 | 14.81 | 4.44 | 13.33 | 22.22 | 10.55 | 20.55 | 30.56 |
| CLoT | 21.67 | 36.67 | 43.33 | 27.33 | 43.83 | 48.33 | 23.00 | 29.33 | 46.67 | 8.00 | 21.10 | 26.67 | 18.66 | 33.33 | 41.67 |
| Ours | **24.06** | **41.75** | **49.59** | **33.65** | **53.57** | **62.19** | **32.92** | **50.52** | **58.53** | **22.74** | **36.18** | **41.58** | **29.62** | **42.37** | **47.42** |
| Improv. | +11.0 | +13.8 | +14.4 | +23.4 | +22.2 | +28.7 | +24.4 | +30.2 | +25.4 | +17.3 | +16.8 | +13.4 | +3.24 | +4.77 | +4.93 |
| **LLaVA-1.5 (7B)** | | | | | | | | | | | | | | | |
| CoT | 1.11 | 1.11 | 1.11 | 1.55 | 2.89 | 3.33 | 1.78 | 2.22 | 2.22 | 1.08 | 3.66 | 5.56 | 7.22 | 11.66 | 13.89 |
| Few-shot | 8.44 | 10.89 | 12.22 | 20.44 | 23.33 | 24.44 | 16.44 | 18.67 | 18.89 | 14.00 | 16.00 | 16.67 | 20.67 | 25.26 | 27.50 |
| Self-consistency | 5.78 | 6.22 | 6.67 | 7.11 | 8.44 | 8.89 | 8.00 | 9.44 | 10.00 | 1.37 | 3.99 | 6.67 | 7.99 | 10.67 | 13.33 |
| HumorousAI | 4.00 | 10.00 | 13.33 | 9.33 | 14.67 | 20.00 | 18.67 | 20.00 | 21.73 | 11.11 | 17.78 | 22.22 | 22.78 | 28.33 | 30.55 |
| LoL | 1.33 | 4.09 | 6.67 | 14.67 | 17.33 | 22.25 | 17.33 | 20.00 | 20.00 | 1.90 | 5.71 | 9.52 | 15.24 | 24.76 | 28.57 |
| Phunny | 3.89 | 10.00 | 13.33 | 15.67 | 22.67 | 24.67 | 13.33 | 18.67 | 22.13 | 4.17 | 13.33 | 22.36 | 10.55 | 20.57 | 27.01 |
| CLoT | 6.40 | 8.00 | 8.00 | 1.8 | 2.40 | 4.00 | 16.00 | 19.60 | 20.00 | 13.33 | 16.11 | 16.67 | 24.44 | 27.78 | 28.98 |
| Ours | **10.22** | **15.22** | **19.56** | **22.89** | **27.67** | **31.11** | **20.67** | **24.44** | **27.67** | **19.33** | **23.17** | **25.00** | **30.16** | **34.33** | **35.83** |
| Improv. | +21.0 | +39.7 | +46.7 | +11.9 | +18.6 | +26.1 | +10.7 | +22.2 | +22.5 | +38.0 | +30.3 | +11.8 | +23.4 | +21.1 | +17.2 |

**Implementation Details.** In the hierarchical imaginator, we impose the top-$k$ relevant jokes $k = 5$ for balancing efficiency and effectiveness, and the threshold of humor-relevant entities $\delta = 5$. For caption generation, all base LLMs use the temperature of 1 to ensure the creative generation of funny captions, leaving all other parameters at their default values. For the humor evaluator, GPT-5 and other humor evaluators use the temperature of 0 to guarantee the stability and reproducibility of evaluation results. Additionally, we fine-tune the Humor-tuned LLaMa3 on ranked caption pairs split 8:1:1 into training, validation, and test sets. Some hyperparameters are analyzed in Section 3.3 and Appendix B.4. More ablation studies and detailed prompt design are provided in Appendix B and G, respectively. All experiments are conducted on 2 NVIDIA RTX 4090 (16 GB) GPUs.

## 3.1 RELIABILITY OF HUMOR EVALUATOR

To measure the reliability of different evaluators, as reported in Table 3, we compare their ranking accuracy in judging human-written caption pairs, which are randomly selected across 200 different contests. Our assessment involves five evaluators: two open-source LLMs with sum token logits and rewards (i.e., LLaMa 3-8B and Humor-tuned LLaMa 3-8B),

Table 3: Ranking accuracy (%) of evaluators.

| Evaluator | Humor in AI | Electronic sheep |
|---|---|---|
| LLaMa 3 | 53.5 | 52.0 |
| Humor LLaMa3 | 60.0 | 58.0 |
| Qwen-Turbo | 55.5 | 54.0 |
| GPT-4.1 | 68.5 | 67.0 |
| GPT-5 | **73.5** | **70.0** |

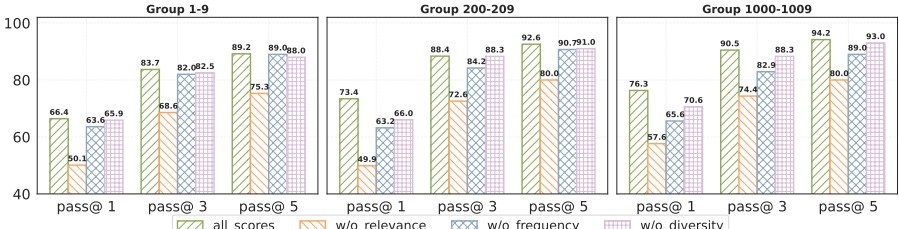

Figure 3: Ablation study of humor-relevance score.

Table 4: Ablation studies of HOMER modules with GPT-4o. Inclusion (✓) or exclusion (✗).

| Module | Image-Only | I+D | I+$\mathcal{C}$ | I+$\mathcal{T}_{\mathrm{im}}$ | I+$\mathcal{C}$+$\mathcal{T}_{\mathrm{im}}$ | I+D+$\mathcal{T}_{\mathrm{im}}$ | I+D+$\mathcal{C}$ | I+D+$\mathcal{C}$+$\mathcal{T}_{\mathrm{im}}$ |
|---|---|---|---|---|---|---|---|---|
| Image ($I$) | ✓ | ✓ | ✓ | ✓ | ✓ | ✓ | ✓ | ✓ |
| Situation Description ($D$) | ✗ | ✓ | ✗ | ✗ | ✗ | ✓ | ✓ | ✓ |
| Conflict Scripts ($\mathcal{C}$) | ✗ | ✗ | ✓ | ✗ | ✓ | ✗ | ✓ | ✓ |
| Imagination Tree ($\mathcal{T}_{\mathrm{im}}$) | ✗ | ✗ | ✗ | ✓ | ✓ | ✓ | ✗ | ✓ |
| Generator | ✓ | ✓ | ✓ | ✓ | ✓ | ✓ | ✓ | ✓ |

| | Humor in AI | | | | | | | | | Electric sheep | | | | | |
| | #Top10 | | | #200-209 | | | #1000-1009 | | | Better Group | | | Worse Group | | |
| Method | @1 | @3 | @5 | @1 | @3 | @5 | @1 | @3 | @5 | @1 | @3 | @5 | @1 | @3 | @5 |
|---|---|---|---|---|---|---|---|---|---|---|---|---|---|---|---|
| Image-Only | 20.20 | 38.30 | 51.00 | 21.00 | 36.00 | 42.99 | 27.99 | 43.30 | 53.00 | 17.67 | 31.33 | 36.67 | 25.67 | 43.50 | 51.67 |
| I+D | 50.60 | 69.49 | 78.00 | 47.20 | 66.50 | 74.00 | 53.60 | 68.60 | 73.00 | 41.66 | 65.16 | 74.99 | 51.50 | 74.00 | 83.33 |
| I+$\mathcal{C}$ | 41.80 | 59.70 | 67.00 | 37.20 | 50.70 | 57.00 | 44.99 | 60.00 | 66.00 | 15.25 | 27.83 | 33.33 | 19.67 | 32.83 | 40.83 |
| I+$\mathcal{T}_{\mathrm{im}}$ | 20.00 | 35.90 | 43.00 | 20.60 | 34.00 | 40.00 | 28.60 | 44.09 | 51.00 | 15.00 | 26.67 | 33.33 | 29.33 | 44.91 | 52.49 |
| I+$\mathcal{C}$+$\mathcal{T}_{\mathrm{im}}$ | 35.40 | 53.89 | 60.00 | 33.00 | 52.60 | 61.00 | 41.00 | 58.70 | 65.99 | 24.00 | 43.00 | 55.00 | 39.17 | 59.50 | 67.50 |
| I+D+$\mathcal{T}_{\mathrm{im}}$ | 34.40 | 56.50 | 67.00 | 36.20 | 51.20 | 56.00 | 42.60 | 59.90 | 67.00 | 36.67 | 57.83 | 68.33 | 51.00 | 72.50 | 80.00 |
| I+D+$\mathcal{C}$ | 57.40 | 75.50 | 80.00 | 56.80 | 74.70 | 79.99 | 63.00 | 78.10 | 83.00 | 60.33 | 74.83 | 78.33 | 68.67 | 82.00 | 86.19 |
| I+D+$\mathcal{C}$+$\mathcal{T}_{\mathrm{im}}$ | **66.41** | **83.70** | **89.18** | **73.40** | **88.38** | **92.57** | **76.32** | **90.50** | **94.19** | **75.53** | **89.21** | **92.10** | **79.45** | **91.48** | **93.81** |

as well as three advanced closed-source LLMs (i.e., Qwen-Turbo, GPT-4.1, and GPT-5). The results indicate that GPT-5 demonstrates superior performance as a humor evaluator. Although Humor-tuned LLaMa 3 shows noticeable improvements, its effectiveness still remains limited. Therefore, we adopt GPT-5 as our primary humor evaluation model. The humor fine-tuning is in Appendix B.7.

## 3.2 FUNNY CAPTION GENERATION

Table 2 demonstrates that our HOMER significantly outperforms seven state-of-the-art baselines on two real-world New York Cartoon Contest datasets, achieving average improvements of 8.62% on pass@1, 6.48% on pass@3, and 5.91% on pass@5 with GPT-4o. Unlike leading methods of multi-modal humor generation, such as HumorousAI and CLoT, which focus on reasoning chains, **HOMER's core distinction is the explicit modeling of the funny caption generation step by step through a humor-theory-driven multi-role framework.** These improvements underscore three key insights: (1) Incorporating humor theory into the generation process provides explicit guidance to LLMs, resulting in captions that are not only more humorous but also more interpretable. In contrast to methods that rely on heuristic reasoning strategies, humor theory enables a systematic, step-by-step generation process, offering greater generation control and interpretability, enhancing the humor quality of the generated captions. (2) Imagination plays a critical role in humor generation. Since humor creation is inherently creative, solely relying on the LLMs' intrinsic imagination may lead to repetitive and limited outputs. By introducing multiple perspectives and diverse imagination patterns, LLMs can generate funnier and more original captions. (3) A multi-role framework facilitates the complex and challenging task of multimodal humor generation by breaking it down into several more precise and refined steps, thereby enhancing the quality of humorous captions.

## 3.3 ABLATION STUDIES

**Ablation on Three Key Modules.** We first exhaust all ablation choices in three humor-theory-driven modules to generate humorous captions in Table 4. Results in Table 4 show that (1) removing any single module consistently degrades performance, verifying the necessity of $D, \mathcal{C}, \mathcal{T}_{\mathrm{im}}$ in the multi-modal caption generation. The largest performance drop, seen in I+D+$\mathcal{T}_{\mathrm{im}}$, highlights the

| Group 1-9 | Group 200-209 | Group 1000-1009 | Group 1-9 | Group 200-209 | Group 1000-1009 |

Figure 4: $k$ hyperparameter.

Figure 5: $\delta$ hyperparameter.

significance of conflict script $\mathcal{C}$ as a basis of caption generation. (2) Both conflict scripts and situation descriptions are critical for deriving imagination trees. Compared to I+$\mathcal{T}_{\text{im}}$, both I+$\mathcal{C}$+$\mathcal{T}_{\text{im}}$ and I+D+$\mathcal{T}_{\text{im}}$ contribute to significant improvements. (3) Inadequate guidance of imagination leads to performance drops, which may lead to irrelevant and nonsensical caption generation. Compared with the image-only variant, I+$\mathcal{T}_{\text{im}}$ leads to performance drops.

**Ablation on humor-relevance score $\mathbf{H}(\cdot)$.** We ablate the calculation of relevance-opposition, frequency, and diversity scores in $\mathbf{H}(\cdot)$ in Figure 3. The *w/o relevance* variant (removing relevance score calculation) consistently results in a significant performance drop, validating the necessity and effectiveness of modeling semantic relevance and conceptual opposition. The *w/o frequency* and *w/o diversity* variants also show a great drop, indicating that they are useful for exploring imagination.

**Robustness of hyperparameters.** We conduct an ablation study on the only two hyperparameters in our method: the number of retrieved jokes $k$ and the number of humor-relevant entities $\delta$, both of which are varied across [3, 5, 7, 9]. The results, as shown in Figures 4 and 5, indicate that our method remains stable across the entire range of values tested, showing strong robustness. We provide the detailed analysis in Appendix B.4.

**Generalization across visual domains.** To assess the generalization ability of our HOMER, we evaluate HOMER on a public ImgFlip meme (Hwang & Shwartz, 2023), which contains a diverse range of images, including realistic, comic, cartoon, and synthetic images. We evaluate the generated meme captions against the ground truth meme captions. Table 5 shows that HOMER consistently outperforms strong competitors by approximately 5.4% on average, validating the effectiveness and powerful generalization ability of our HOMER across different visual domains. More results are in Appendix Table 9.

Table 5: Results on Meme data(%).

| Method | pass@1 | pass@3 |
|---|---|---|
| CoT | 74.33 | 86.12 |
| Few-Shot | 66.67 | 81.67 |
| Self-consistency | 70.00 | 90.83 |
| HumorousAI | 75.00 | 80.00 |
| LoL | 71.67 | 81.12 |
| Phunny | 21.67 | 31.67 |
| CLoT | 76.67 | 88.33 |
| **HOMER** | **83.33** | **96.67** |

## 3.4 CASE STUDY

We show two cases in Figure 6, showing the explicit intermediate results of caption generation. For **Case 1**, the extractor records the core opposition, *normal coffee cups vs. gigantic cups*. The hierarchical imaginator then expands a traceable imagination path from the chosen target *coffee cups* to milk→cream→cow, supported by retrieval. Finally, the GTVH-guided generator generates ***HR says we can expense a cow now***. This caption suggests that employees, drinking large quantities of coffee, humorously claim HR allows them to expense a whole cow for milk. The exaggeration aligns with both the image and the office culture. For **Case 2**, the core script opposition is *communication from a typical col-*

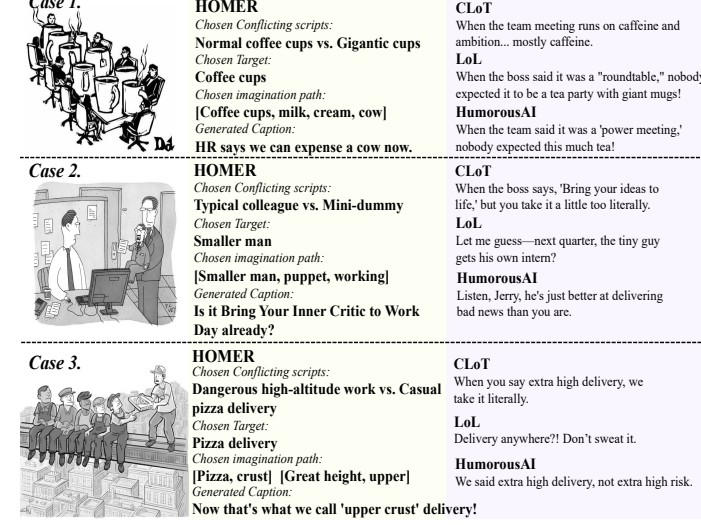

Figure 6: Case Study.

*league vs. through a mini-dummy* as the fundamental joke logic. The hierarchical imaginator derives smaller man→puppet→working imagination chain of humor target *Smaller man*. Finally, the GTVH-guided generator generates the funny caption ***Is it Bring Your Inner Critic to Work Day already?*** that fuses idiom subversion with personification to resolve the visual incongruity in a fresh way. For **Case 3**, which is a normal image but with incongruity, the core script opposition is the dangerous high-altitude work vs. the casual pizza delivery, forming the basis of the joke. The hierarchical imaginator derives two imagination chains: great height→upper and pizza→crust. Then, the generator derives the funny caption ***Now that's what we call 'upper crust' delivery!***. This caption cleverly plays on the idiom "upper crust," connecting the visual context of a pizza being delivered at a great height to the phrase's meaning of high quality or elite status.

### 3.5 HUMAN EVALUATION

We conduct a human evaluation in which 20 raters scored seven captions corresponding to seven methods on a five-point funniness rating rule (1: not funny, 2: slightly funny, 3: moderately funny, 4: funny, 5: very funny), following standard caption evaluation practice (Kasai et al., 2022; Levinboim et al., 2021). There are a total of 5600 rating scores for evaluation. Table 6 reports the mean ratings ($\pm$ std) for seven representative methods. Our method achieves

Table 6: Mean human ratings with std ($\pm$).

| Method | Humor in AI | Electronic sheep |
|---|---|---|
| CoT | 2.47$\pm$ 0.67 | 2.20$\pm$ 0.78 |
| Few-shot | 2.96$\pm$ 0.70 | 2.56$\pm$ 1.00 |
| Self-consistency | 2.66$\pm$ 0.59 | 2.25$\pm$ 0.56 |
| CLoT | 2.95$\pm$ 0.77 | 2.53$\pm$ 0.73 |
| HumorousAI | 3.01$\pm$ 0.73 | 2.24$\pm$ 0.81 |
| LoL | 3.16$\pm$ 0.84 | 2.40$\pm$ 0.82 |
| Ours | **3.54** $\pm$ 0.59 | **3.31**$\pm$ 0.85 |

the highest mean score ($> 3.0$ on the five-point funniness scale), indicating that human raters generally judged its captions to be over moderately funny. Inter-rater agreement is relatively substantial, with Cohen's kappa $\kappa = 0.49$, following agreement measurements in human studies (Hallgren, 2012). Humor is inherently subjective, as individuals differ in their interpretations of humor as well as in their understanding of images. Therefore, $\kappa = 0.49$ reflects an acceptable level of agreement among annotators, given the expected subjectivity in humor evaluation tasks. Detailed human evaluation can be found in Appendix B.8.

## 4 RELATED WORK

**Humor creativity in LLMs.** With the emergence of LLMs, exploring the linguistic capability of LLMs on human-challenging linguistic phenomena (Shang & Huang, 2025; Sun et al., 2025; Zhou et al., 2025a), such as multi-modal humor generation, has attracted rapidly growing interest from researchers (Horvitz et al., 2024; Cocchieri et al., 2025; Baluja, 2025; Wang et al., 2025; Attardo, 2024). Benchmark evaluations show that prominent models (e.g., GPT-4 variants) can detect surface humor cues yet struggle with originality and comedic quality (Zhang et al., 2024; Wu et al., 2025). To improve the humor generation ability of LLMs, prior methods typically rely on generic prompting (Zhang et al., 2024; Chen et al., 2024), multi-hop reasoning for self-improvement (Zhong et al., 2024), or task-specific tuning (Wang et al., 2025) to better steer model outputs towards funnier captions. However, they still suffer limitations of interpretability and creativity. Therefore, we propose a humor generation mechanism driven by humor theory and augmented by a hierarchical creative imagination process. Classical related works can be found in Appendix F.

## 5 CONCLUSIONS

In this paper, we propose a HOMER humor generation framework to address the limitations of interpretability and creativity in prior approaches. Anchored in the famous theory GTVH, HOMER employs three coordinated roles: a conflict-script extractor that identifies detailed situation descriptions and script oppositions, a hierarchical imaginator that stimulates imaginative associations with retrieval, and a caption generator that generates funny captions using the obtained knowledge resources. This modular design shows explicit control over humor logic and materials, enabling targeted editing and more original creativity. Extensive experiments demonstrate consistent improvements over seven state-of-the-art baselines for multimodal humor captioning.

ACKNOWLEDGMENTS

This work is supported by grants from the Research Grants Council of the Hong Kong Special Administrative Region, China (No. 12201923).

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

---

**Algorithm 1** HOMER Framework.

---

**Require:** A cartoon image $I$, the number of top-$k$ relevant jokes $k$, the threshold of retrieved entities rank $\delta$.
**Ensure:** Generated funny cartoon captions $\text{Cap}(I)$.

1: **Phase I: Script Extraction:** $(\mathcal{C}, D) \leftarrow \text{Extract}(I)$;   $\triangleright \mathcal{C}$: Script Oppositions, $D$: Situation Description
2:   $D \leftarrow \text{Extract}(I)$;
3:   Define script opposition $\Phi_{\text{scripts}}(\cdot)$ and design a prompt as $\Phi_{\text{scripts}}(I, D)$;
4:   $\mathcal{C} \leftarrow \text{Extract}(\Phi_{\text{script}}(I, D))$;
5: **Phase II: Imagination:** $\mathcal{T}_{\text{im}} \leftarrow \text{Imagine}(I, \mathcal{C}, D)$;
6:   **Choose candidate targets from local and global perspectives:**
7:    $V = \{\text{loc,glob}\}, T_{\text{root}} = \emptyset$;
8:    $O_{\text{loc}} \leftarrow D, O_{\text{glob}} \leftarrow I$;   $\triangleright$ Local: fine-grained situation description. Global: obvious scene entities
9:    **for** $\forall v \in V$ **do**
10:     $\text{Ent}(O_v, \mathcal{C}) = \{t_1, ..., t_m\}, T_{\text{root}} \leftarrow T_{\text{root}} \cup \text{Ent}(O_v, \mathcal{C})$;
11:   **Deep-pattern imagination forms backbone chains:**
12:    **for** $\forall t_i \in T_{\text{root}}$ **do**
13:     $T'_i \leftarrow \langle t_i \rangle, e_0^{(i)} \leftarrow t_i$;             $\triangleright$ Initialize chain by $t_i$
14:     **for** $\forall \tau \in \{0, ..., n-1\}$ **do**          $\triangleright$ $n$ is determined by LLMs
15:      $e_{\tau+1}^{(i)} = f_{\text{chain}}(e_\tau^{(i)}), \quad T'_i \leftarrow T'_i + \langle e_{\tau+1}^{(i)} \rangle$;
16:    $\mathcal{T}' = \{T'_i | t_i \in T_{\text{root}}\}$;
17:    $\mathcal{T} = \{T_i | t_i \in T_{\text{root}}\} \leftarrow f_{\text{merge}}(\mathcal{T}')$;     $\triangleright$ Align local and global entities across $\mathcal{T}'$
18:   **Broad-pattern imagination through humor-based retrieval:**
19:    **for** $\forall t_i \in T_{\text{root}}, \forall e_\tau^{(i)} \in T_i$ **do**
20:     $\mathbf{z}_q = f_{\text{emb}}(D, \mathcal{C}, e_\tau^{(i)})$;
21:     **for** $\forall j \in \mathcal{J}$ **do**
22:      $sim(\mathbf{z}_q, \mathbf{z}_j) \leftarrow \frac{\mathbf{z}_q \cdot \mathbf{z}_j}{||\mathbf{z}_q|| ||\mathbf{z}_j||}$;
23:     $J_{\text{topK}} = \underset{j \in \mathcal{J}}{\text{argtopk}}(sim(\mathbf{z}_j, \mathbf{z}_q))$;
24:     **for** $j \in J_{\text{topK}}$ **do**
25:      $\mathcal{E}_j \leftarrow \{\varepsilon_1, \varepsilon_2, ...\}$ from $j$;       $\triangleright$ Tokenize and lemmatize the joke $j$
26:      **for** $\forall \varepsilon \in \mathcal{E}_j$ **do**
27:       $\mathcal{H}_{\text{rel}}(e_\tau^{(i)}, \varepsilon) = \text{TSS}(s_{e_\tau}, s_\varepsilon) + f(\text{TSS}(s_{e_\tau}, s_\varepsilon))CO(s_{e_\tau}, s_\varepsilon)$;   $\triangleright$ Relevance score
28:       $\mathcal{H}_{\text{freq}}(\varepsilon) = \sqrt{\frac{\sum_{j \in J_{\text{topK}}} \text{count}(\varepsilon, \mathcal{E}_j)}{\sum_{j \in J_{\text{topK}}} |\mathcal{E}_j|} \frac{\sum_{j \in J_{\text{topK}}} \mathbf{1}[\varepsilon \in \mathcal{E}_j]}{|J_{\text{topK}}|}}$;   $\triangleright$ Frequency score
29:       $\mathcal{H}_{\text{div}}(\varepsilon)) = \frac{\sum_{p \in P} \mathbf{1}[N(\varepsilon) > 0]}{|P|}$;      $\triangleright$ Diversity score
30:       $\mathcal{H}(e_\tau^{(i)}, \varepsilon) = \mathcal{H}_{\text{rel}}(e_\tau^{(i)}, \varepsilon) + \mathcal{H}_{\text{freq}}(\varepsilon) + \mathcal{H}_{\text{div}}(\varepsilon)$;
31:      **Prune:** $\mathcal{E}_{\text{leaf}} \leftarrow \{\varepsilon | \text{rank}(\mathcal{H}(e_\tau^{(i)}, \varepsilon)) \leq \delta\}$;
32:     $T_i \leftarrow T_i \cup \{(e_\tau^{(i)}, \varepsilon) | \varepsilon \in \mathcal{E}_{\text{leaf}}\}, \mathcal{T}_{\text{im}} \leftarrow \{T_i | t_i \in T_{\text{root}}\}$;
33: **Phase III: Generation:** $\text{Cap}(I) \leftarrow \text{Gen}(\mathcal{C}, D, \mathcal{T}_{\text{im}}, \Omega)$;
34:   $C \leftarrow \text{Sample}(\mathcal{C}), t_i \leftarrow \text{SelectTargets}(T_{\text{root}}, C)$; $\triangleright$ Randomly select conflict scripts and relevant targets
35:   $\mathcal{P}_i \leftarrow \text{Path}(T_i)$ by DFS, $T_i \in \mathcal{T}_{\text{im}}$;     $\triangleright$ Enumerate all paths from ancestor to leaf;
36:   $P_i \leftarrow \text{SamplePath}(\mathcal{P}_i)$;         $\triangleright$ Randomly sample one imagination path
37:   $\Phi(\mathcal{C}, D, P_i, \Omega)$;           $\triangleright$ Construct the GTVH-guided prompt.
38:   $\text{Cap}(I) \leftarrow \text{Gen}(\Phi(\mathcal{C}, D, P_i, \Omega))$;
39: **return** $\text{Cap}(I)$;

---

# A   HOMER ALGORITHM.

We propose HOMER, a three-phase framework for humorous image captioning, summarized in Algorithm 1. Phase I (lines 1–4) extracts core conflicting scripts via an extractor. Phase II (lines 5–32) expands humorous imagination with a hierarchical imaginator by (i) initializing candidate humor targets from local and global views guided by the conflicting scripts (lines 5–10), (ii) performing deep-pattern imagination via LLM-driven associations to form free-association backbone chains (lines 11–17), and (iii) conducting humor-relevance retrieval to grow the chains into imagination trees (lines 18–32). Phase III (lines 33–39) employs a GTVH-guided generator to generate funny captions conditioned on five constructed knowledge resources.

# B EXPERIMENTS

## B.1 DATASETS

**Humor Retrieval Database Construction.** In particular, we collect and reorganize 11 humor benchmarking datasets as our humor retrieval dataset, which are from Pun of a Day (Yang et al., 2015), Short Jokes (Annamoradnejad & Zoghi, 2020), Reddit Jokes (Weller & Seppi, 2019), rJoke (Weller & Seppi, 2020), SemEval 2021 Task 7 (Meaney et al., 2021), TED Laughter (Kim, 2014), HumorNorm (Engelthaler & Hills, 2018), CleanComedy (Vikhorev et al., 2024), ShortJokes[1], CrowdTruth[2] and Dad Jokes[3].

**Multi-stage data curation.** We construct our humor retrieval database through a multi-stage data curation process. First, we collect several publicly available humor-related datasets. Next, as an initial filtering step, we employ humor rating information provided within the datasets to eliminate entries rated as not funny; specifically, all jokes with a humor rating lower than 3 are discarded. Subsequently, we perform data cleaning to remove noise and ensure quality. To further refine the corpus, we eliminate duplicate jokes as well as jokes that exhibit excessive textual similarity. In particular, for any pair of jokes sharing more than 80% of their English words, we retain only the longer version. After completing these operations, our finalized humor retrieval database comprises a total of 335,570 jokes.

**Comparison of joke database and our test dataset.** The joke database consists primarily of one-liner text jokes, explicitly excluding cartoons or image captions, whereas the test set comprises original, publicly submitted captions from the New Yorker Caption Contest. The database is used exclusively for text-only humor tasks (humor detection, rating, and joke generation), while the test set is reserved for multimodal humor tasks, specifically funny caption generation, ensuring a clear separation of modalities and application contexts. A summary of formats, sources, and task usage is provided in Table 7.

**License and curation policy:** All datasets used are publicly available and we follow a strict curation protocol to prevent cross-modal leakage: (i) we exclude any datasets that are multimodal or have been used for multimodal applications; (ii) the corpus is restricted to text-only humor, focusing on short, one-liner jokes with simple structure; and (iii) we will remove items that are near-duplicates or overly similar to content in our multimodal captioning evaluation, minimizing any risk of overlap.

Table 7: Overview of humor-related corpora

| Corpus | Data Format | Source(s) | Intended Use |
|---|---|---|---|
| Short Jokes | One-liner jokes | Various joke websites | Text-only humor |
| Reddit Jokes | One-liner jokes | Reddit (r/Jokes) | Text-only humor |
| Pun of the Day | One-liner jokes | Pun of the Day website | Text-only humor |
| rJoke | One-liner jokes | Reddit (r/Jokes) | Text-only humor |
| SemEval 2021 Task 7 | One-liner jokes | SemEval 2021 Task 7 | Text-only humor |
| TED Laughter | Speech | TED Talks | Text-only humor |
| HumorNorm | Words | English lexical resources | Text-only humor |
| CleanComedy | One-liner jokes | Reddit, Twitter, other platforms | Text-only humor |
| CrowdTruth | One-liner jokes | Various joke websites | Text-only humor |
| Dad Jokes | One-liner jokes | Grin, Dad Joke It | Text-only humor |
| Our Test Dataset | Cartoons with captions | New Yorker Caption Contest | Multimodal humor |

**Data distribution of tested datasets.** Our method generalizes beyond images with overt anomalies, effectively handling a diverse range of humor sources, including unexpected logic, contextual incongruity, personification, and role reversals. Its foundation is *script opposition*, a central concept in the General Theory of Verbal Humor (GTVH) and related accounts, where humor emerges from

---

[1]https://github.com/amoudgl/short-jokes-dataset
[2]https://github.com/CrowdTruth/Short-Text-Corpus-For-Humor-Detection
[3]https://www.kaggle.com/datasets/usamabuttar/dad-jokes

surprising conflicts between competing scripts. We statistic different types of script opposition in the dataset across the following dimensions:

- Abnormalities: visual elements that deviate from everyday norms.
- Unexpected logic: outcomes that defy conventional expectations (e.g., role reversals).
- Contextual incongruity: entities or actions that are inconsistent with their context.
- Exaggeration: phenomena or behaviors presented in an extreme form.
- Ambiguity: multiple plausible interpretations that invite playful confusion.
- Personification: nonhuman entities endowed with human traits, intentions, or roles.

Table 8: Data distribution in the Humor in AI dataset

| Humor Basis | Occurrences | Percentage |
|---|---|---|
| Abnormalities | 122 | 34% |
| Unexpected Logic | 70 | 19% |
| Contextual Incongruity | 67 | 18% |
| Exaggeration | 51 | 14% |
| Ambiguity | 55 | 15% |
| **Total** | **365** | **100%** |

## B.2 COMPETITORS

Recent advances in multimodal and language-based humor generation have led to the development of several benchmark methods. **HumorousAI**(Zhang et al., 2024) and **CLoT**(Zhong et al., 2024) represent state-of-the-art approaches for multimodal humor generation, typically leveraging advanced reasoning strategies to create contextually appropriate and funny captions. In contrast, **LoL**(Wang et al., 2025) addresses dialogue-based humor generation through a specialized fine-tuning framework tailored to conversational contexts. Additionally, **Phunny**(Chen et al., 2024) focuses on the generation of puns, specifically targeting linguistic wordplay and double meanings.

## B.3 METRICS

**Unbiased Pass@$k$ Calculation.** To evaluate the performance of our method, we employ the unbiased pass@$k$ metric (Chen et al., 2021), which estimates the probability that at least one of $k$ randomly selected captions is funnier than the human-written caption. For a given image, we sample $n$ candidate captions from the model, and let $c$ denote the number of captions that are funnier than the ground truth human caption among them. The unbiased pass@$k$ for this task is computed as $1 - \frac{\binom{n-c}{k}}{\binom{n}{k}}$, where $\binom{n}{k}$ denotes the binomial coefficient. This formula corrects for the bias that may arise when multiple winner captions exist among the samples. Averaging over all $N$ images in the dataset, the overall unbiased pass@$k$ metric is calculated as

$$\text{pass@k} = \frac{1}{N} \sum_{i=1}^{N} \left[ 1 - \frac{\binom{n_i-c_i}{k}}{\binom{n_i}{k}} \right], \tag{8}$$

where $n_i$ is the total number of generated captions for the $i$-th image. We set $n_i = 5$ in our experiments, and $c_i$ is the number of captions in $k$ sampled captions judged to be funnier than the corresponding human caption by the evaluator. This unbiased estimation provides a reliable measure of the model's win rate given multiple sampling trials.

## B.4 HYPERPARAMETER ANALYSIS

**Analysis of $k$ and $\delta$ in the joke database retrieval.** Optimal performance is observed when $k$ is set to 3, 5, or 7 and when $\delta$ is set to 5 or 7, suggesting that retrieving too few humor instances results in an insufficient imagination space, whereas retrieving too many introduces noise, such as unrelated

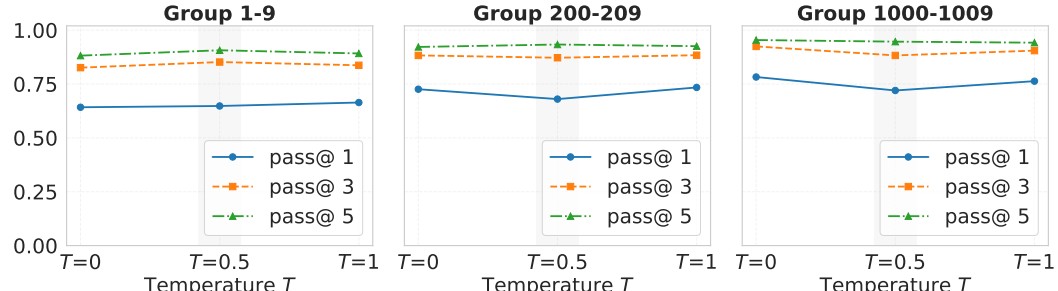

Figure 7: Robustness of the LLM temperature.

Table 9: Performance comparison on Meme dataset with pass@k metrics

| Method | pass@1 | pass@3 | pass@5 |
|---|---|---|---|
| CoT | 74.33 | 86.12 | 95.83 |
| Few-Shot | 66.67 | 81.67 | 91.67 |
| Self-consistency | 70.00 | 90.83 | 91.67 |
| HumorousAI | 75.00 | 80.00 | 83.33 |
| LoL | 71.67 | 81.12 | 97.54 |
| Phunny | 21.67 | 31.67 | 33.33 |
| CLoT | 76.67 | 88.33 | 91.67 |
| **HOMER** | **83.33** | **96.67** | **98.86** |

entities, which can degrade performance. Therefore, we select $k = 5$ and $\delta = 5$ in our method to strike a balance between maintaining high performance and minimizing the noisy inducing.

**Robustness to Temperature Variation.** We evaluate the stability of our method under different sampling temperatures of the LLM by varying the temperature parameter across the values $0$, $0.5$, and $1$, as shown in Figure 7. Experimental results demonstrate that our approach achieves consistently strong performance across all tested temperature settings, indicating robustness to changes in the sampling temperature. The results suggest that the effectiveness of our method is not sensitive to the choice of temperature within this range, underscoring its practical reliability and generalizability in diverse decoding scenarios.

### B.5 COMPARISON OF HUMOR FREQUENCY SCORE AND TF-IDF

The distinction between TF-IDF and our humor-relevance metric lies in their weighting schemes: TF-IDF down-weights globally common tokens, whereas our approach up-weights tokens and humor concepts that recur in jokes relevant to a given situation and its conflicting scripts. We aim to identify entities that are both salient and frequently used as reliable components for humor generation. The objectives of the two methods are opposite. In experiments, our humor-frequency score outperformed a TF-IDF-based baseline, as shown in Table 10.

Table 10: Comparison of scoring methods on pass@k

| Scoring Method | pass@1 | pass@3 | pass@5 |
|---|---|---|---|
| **TF-IDF** | 63.00 | 79.83 | 86.33 |
| **Humor-Frequency (Ours)** | 66.41 | 83.70 | 89.18 |

### B.6 EVALUATION ON FOUR DIMENSIONS

We conduct comprehensive qualitative and quantitative analyses across four key dimensions of multi-modal humor generation, leveraging both expert LLMs (GPT-5) and automated metrics: **(1) visual understanding**, **(2) humor understanding**, **(3) humor imagination**, and **(4) stylistic expression**. The results are shown in Table 11 and Table 12.

For the dimensions requiring nuanced linguistic and cognitive judgment—*visual understanding, humor understanding,* and *stylistic expression*—we use GPT-5 as an expert evaluator. Specifically, for each image, GPT-5 ranks the humorous captions generated by eight different methods according to well-defined criteria for each dimension, assigning ranks from best (#1) to worst (#8). The average rank for each method (lower is better) is reported to ensure a fair and consistent assessment. To enhance reliability, each ranking is conducted over five repeated trials. For the dimension of *humor imagination*, we utilize two established automated metrics: (i) **n-gram diversity**, which quantifies lexical variety, and (ii) **NLI diversity**, measuring the percentage of non-entailing caption pairs as judged by a state-of-the-art Roberta-large natural language inference model. Higher scores on these metrics indicate a greater capacity for imagination, as producing a wider variety of humorous captions reflects the model's ability to generate diverse and creative comedic ideas.

Table 11: Humor in AI Dataset: Comparative results across four dimensions

| Model | Visual Understanding Avg. Rank ($\downarrow$) | Humor Understanding Avg. Rank ($\downarrow$) | Stylistic Expression Avg. Rank ($\downarrow$) | Humor Imagination | |
|---|---|---|---|---|---|
| | | | | 3-gram ($\uparrow$) | NLI Diversity (%) ($\uparrow$) |
| CoT | 5.5 | 4.8 | 5.6 | 0.87 | 85.3 |
| Few-Shot | 4.4 | 3.7 | 3.8 | 0.84 | 81.6 |
| Self-Consistency | 5.9 | 5.4 | 5.7 | 0.88 | 85.9 |
| HumorousAI | 4.1 | 4.9 | 4.5 | 0.59 | 70.0 |
| LoL | 4.9 | 4.6 | 4.4 | 0.92 | 89.3 |
| Phunny | 5.9 | 5.6 | 5.4 | 0.69 | 81.9 |
| CLoT | 4.5 | 3.8 | 4.3 | 0.46 | 51.5 |
| **HOMER (Ours)** | **2.5** | **3.2** | **2.3** | **0.98** | **91.5** |

Table 12: Electronic Sheep Dataset: Comparative results across four dimensions

| Model | Visual Understanding Avg. Rank ($\downarrow$) | Humor Understanding Avg. Rank ($\downarrow$) | Stylistic Expression Avg. Rank ($\downarrow$) | Humor Imagination | |
|---|---|---|---|---|---|
| | | | | 3-gram ($\uparrow$) | NLI Diversity (%) ($\uparrow$) |
| CoT | 3.8 | 4.3 | 4.2 | 0.88 | 84.2 |
| Few-Shot | 4.5 | 4.5 | 4.1 | 0.89 | 83.1 |
| Self-Consistency | 4.2 | 4.5 | 4.8 | 0.84 | 86.40 |
| HumorousAI | 6.9 | 6.7 | 6.1 | 0.92 | 88.5 |
| LoL | 4.5 | 4.6 | 5.7 | 0.94 | 91.9 |
| Phunny | 6.8 | 6.8 | 6.9 | 0.89 | 85.2 |
| CLoT | 3.5 | 3.2 | 2.7 | 0.83 | 78.9 |
| **HOMER (Ours)** | **1.8** | **1.4** | **1.5** | **0.98** | **92.2** |

### B.7 TWO-STAGE HUMOR TUNING STRATEGY OF HUMOR EVALUATOR

For the Humor-tuned LLaMa 3, we utilize a two-stage training strategy: supervised fine-tuning (SFT) followed by Direct Preference Optimization (DPO). This process encourages the model to align with human preferences for humor by assigning higher rewards to funnier and lower rewards to less funny captions in the benchmarking ranking. During inference, Humor-tuned LLaMa 3 assigns a reward score to each caption. We then assess whether the model can correctly predict the ground-truth ranking by verifying that the higher reward corresponds to the higher rank.

### B.8 HUMAN EVALUATION.

**Procedure of human evaluation.** Human evaluation of caption funniness is conducted using a standardized procedure. As shown in Figure 8, for each cartoon image, human annotators are presented with seven candidate captions and asked to rate the funniness of each caption on a scale from 1 to 5. The assessment proceeds in two steps: first, evaluators must decide whether the caption is relevant to the given cartoon. If the caption is deemed irrelevant, it automatically receives a score of 1.0. If the caption is relevant, annotators then assess its comedic quality according to the following scale: Not Funny (1.0), Slightly Funny (2.0), Moderately Funny (3.0), Funny (4.0), and Very Funny (5.0). This protocol ensures that both relevance and humor are systematically appraised and provides a

Table 13: Human evaluation of seven methods: Mean, Standard Deviation (SD), and Median of Humor Ratings by 20 Raters (1-5 Scale).

| Dataset | Humor in AI | | | Electronic sheep | | |
|---------|------|------|--------|------|------|--------|
| Method | Mean | SD | Median | Mean | SD | Median |
| CoT | 2.47 | 0.6697 | 2.50 | 2.20 | 0.7756 | 2.45 |
| Few-shot | 2.96 | 0.6992 | 2.80 | 2.56 | 1.0065 | 3.10 |
| CLoT | 2.95 | 0.7732 | 2.75 | 2.53 | 0.7314 | 2.60 |
| Self-consistency | 2.66 | 0.5949 | 2.60 | 2.25 | 0.5520 | 2.40 |
| HumorousAI | 3.01 | 0.7301 | 2.75 | 2.24 | 0.8093 | 1.85 |
| LoL | 3.16 | 0.8338 | 3.40 | 2.40 | 0.8236 | 2.15 |
| Ours | 3.54 | 0.5862 | 3.65 | 3.31 | 0.8491 | 3.20 |

fine-grained numeric measure of each caption's effectiveness in eliciting amusement. The human evaluation was conducted on a total of 2,800 data points, calculated as 20 cartoon images were evaluated by 20 human raters across 7 different caption generation methods (20 images × 20 raters × 7 methods).

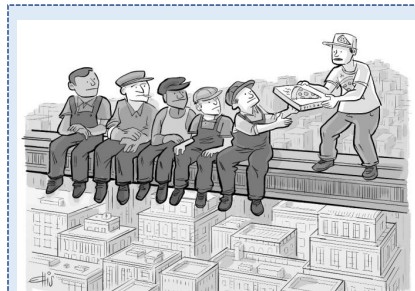

**Example of Human Evaluation**

Instructions

Based on the cartoon image, rate the funniness of the six cartoon captions (1-5). First, determine whether it is relevant. If not, score 1.0. If relevant, score the funniness.

Not Funny (1.0). Slightly Funny (2.0). Moderately Funny (3.0). Funny (4.0). Very Funny (5.0).

Figure 8: Example of human evaluation.

**Detailed human evaluation.** The results of our human evaluation, summarized in Table 13, demonstrate that our method consistently outperforms six baseline methods across two humor datasets, "Humor in AI" and "Electronic Sheep." 20 human raters scored each method using a 1-5 scale. Our approach achieves the highest mean humor ratings on both datasets. These results collectively indicate the robustness and effectiveness of our approach in generating humorous content as perceived by human evaluators.

### B.9 DIVERSITY EVALUATION.

We assess caption diversity using two established metrics. The results are shown in Tables 14 and 15.

- *N-gram (Distinct-N) diversity*: a lexical-variability measure computed as the ratio of unique $n$-grams to the total number of generated $n$-grams (typically for $n \in \{1, 2, 3\}$).

- *NLI Diversity*: the percentage of caption pairs classified as *non-entailing* by a widely adopted *RoBERTa-large* natural language inference model, capturing semantic variety beyond surface form.

Higher scores on either metric indicate a more diverse set of captions. Across both metrics, our results provide strong empirical evidence that *HOMER* generates more diverse funny captions, consistently outperforming all baselines.

### B.10 CASE STUDIES

Table 14: Humor in AI Dataset: n-gram coverage and NLI diversity

| Model | 1-gram (↑) | 2-gram (↑) | 3-gram (↑) | NLI Diversity (↑) |
|---|---|---|---|---|
| HumorousAI | 0.45 | 0.55 | 0.59 | 70.0% |
| LoL | 0.64 | 0.83 | 0.87 | 89.3% |
| Phunny | 0.50 | 0.64 | 0.69 | 81.9% |
| CLoT | 0.35 | 0.43 | 0.46 | 51.5% |
| **Our Method** | **0.76** | **0.94** | **0.98** | **91.5%** |

Table 15: Electronic Sheep Dataset: n-gram coverage and NLI diversity

| Model | 1-gram (↑) | 2-gram (↑) | 3-gram (↑) | NLI Diversity (↑) |
|---|---|---|---|---|
| HumorousAI | 0.65 | 0.87 | 0.92 | 88.5% |
| LoL | 0.63 | 0.88 | 0.94 | 91.9% |
| Phunny | 0.64 | 0.84 | 0.89 | 85.2% |
| CLoT | 0.58 | 0.78 | 0.83 | 78.9% |
| **Our Method** | **0.72** | **0.94** | **0.98** | **92.2%** |

In Figure 9, the conflict between a professional office setting with unexpected horseplay juxtaposes scripts of routine and hyperbole, yielding a humorous reading. This script opposition leverages surprise, incongruity, and cognitive resolution, which are central to effective and engaging humor.

### B.11 HARMFUL DETECTION.

We evaluate harmful content in HOMER's generated captions using Detoxify(Hanu & Unitary team, 2020), a widely used toxicity detector, across seven dimensions: toxicity, severe toxicity, obscene, identity attack, insult, threat, and sexual explicit, as shown in Figure 10. On *Humor in AI*, average scores of the dataset in seven dimensions are very low, summing to 0.023 ($<$ 0.03). On *Electronic Sheep*, the sum of toxicity is 0.015 ($<$ 0.02). These consistently low scores indicate negligible harmful content, suggesting that our HOMER generates captions that largely avoid abusive, threatening, or sexually explicit language.

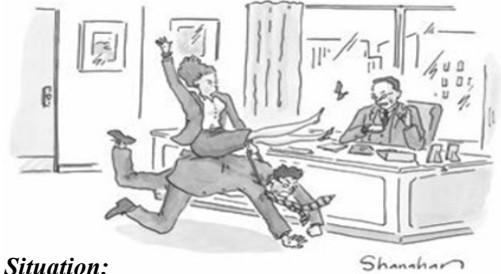

***Situation:***
In the office with professional colleagues

***Script opposition:***
Professional behavior vs. unexpected horseplay

| ***Target:*** | ***Narrative strategy:*** | ***Language:*** |
|---|---|---|
| Cowgirl | Short narrative | Wordplay |

***Funny caption generated by HOMER:***
Well, Janet, I admire your drive, but this isn't exactly what we meant by *taking the reins at work*.

Figure 9: GTVH-guided Example.

We also evaluate more harmful content in captions generated by HOMER using Detoxify (Hanu & Unitary team, 2020), as shown in Figure 11. Specifically, harmfulness is assessed across seven dimensions: toxicity, severe toxicity, obscene language, identity attack, insult, threat, and sexual explicitness. This evaluation is performed on two datasets, Humor in AI and Electronic Sheep, and covers captions produced by three base LLMs: Claude-4, Qwen-VL, and LLaVA. The results consistently demonstrate that harmful content scores are very low across all toxicity dimensions, datasets, and base models. These findings indicate that HOMER reliably generates captions that are safe, minimizing the risk of toxic, offensive, or otherwise inappropriate outputs in diverse settings.

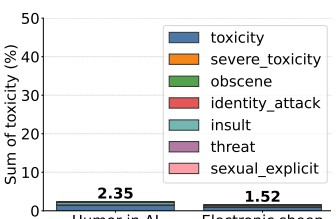

Figure 10: Harmful detection.

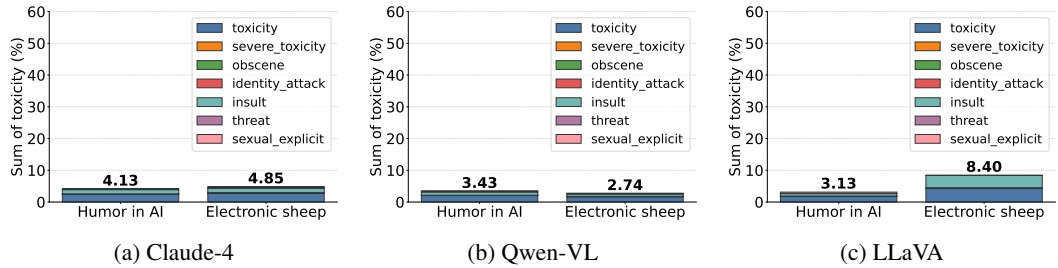

Figure 11: Harmful detection of three base models.

## B.12 COMPUTATIONAL COSTS

**Efficiency of HOMER.** Table 16 presents the number of API calls required at each stage of HOMER for both the full model and the naive model. The extractor and generator stages make the same number of API calls in both models. However, in the naive model, the imaginator stage does not make any API calls because the imagination results of LLM association can be pre-processed and saved in advance. Humor-retrieval does not need API calls. In contrast, the full model involves three API calls at the imaginator stage, reflecting real-time imagination processing. This comparison highlights the efficiency gained by pre-processing imagination results in the naive model.

Table 16: Number of API calls for each stage of HOMER

| Model | Full Model | Naive Model |
|---|---|---|
| Extractor | 2 | 2 |
| Imaginator | 3 | 0 |
| Generator | 2 | 2 |

**Fair comparison under the same LLM calls.** For fairness, each baseline generates multiple independent humorous captions and selects the best one for each output. The number of attempts was set to match or exceed HOMER's seven LLM calls. We show the pass@1 results evaluated by GPT-5 as Table 17. Despite with equal or greater call budgets, baselines show modest gains and do not close the performance gap with HOMER, indicating that HOMER's superiority is from its collaborative, structured framework rather than a higher computational budget.

Table 17: LLM calls per output and pass@1 performance across methods.

| Method | LLM Calls per Output | pass@1 (%) |
|---|---|---|
| HumorousAI | 9 (=3 calls × 3 repeats) | 62.7 |
| LoL | 8 (=4 calls × 2 repeats) | 58.0 |
| Phunny | 9 (=3 calls × 3 repeats) | 19.1 |
| CLoT | 7 (=7 calls × 1 repeats) | 61.2 |
| **HOMER (Ours)** | **7** | **66.4** |

## B.13 SIGNIFICANCE TEST AND AGREEMENT EVALUATION

We assess statistical significance with a two-sided Wilcoxon signed-rank test on Pass@k scores and quantify agreement between GPT-5 and human judgments via correlation analysis. The results show that our method significantly outperforms all baselines ($p < 0.05$), as shown in Table 18

We measure the agreement between GPT-5 scores and human ratings per caption via the Pearson correlation coefficient. The results in Table 19 show that GPT-5 and humans are positively strongly correlated.

Table 18: Pairwise significance test results for Humor in AI (left) and Electronic Sheep (right) datasets ($\alpha = 0.05$).

| Comparison Pair | p-value | Significant? |
|---|---|---|
| **Ours** vs. CoT | 0.0000 | Yes |
| **Ours** vs. Few-Shot | 0.0027 | Yes |
| **Ours** vs. Self-Consistency | 0.0031 | Yes |
| **Ours** vs. CLoT | 0.0153 | Yes |
| **Ours** vs. HumorAI | 0.0036 | Yes |
| **Ours** vs. LoL | 0.0001 | Yes |
| **Ours** vs. Phunny | 0.0000 | Yes |

**Humor in AI dataset**

| Comparison Pair | p-value | Significant? |
|---|---|---|
| **Ours** vs. CoT | 0.0000 | Yes |
| **Ours** vs. Few-Shot | 0.0120 | Yes |
| **Ours** vs. Self-Consistency | 0.0001 | Yes |
| **Ours** vs. CLoT | 0.0011 | Yes |
| **Ours** vs. HumorAI | 0.0001 | Yes |
| **Ours** vs. LoL | 0.0004 | Yes |
| **Ours** vs. Phunny | 0.0000 | Yes |

**Electronic Sheep dataset**

Table 19: Pearson correlation coefficients by evaluator pair

| Evaluator Pair | Pearson coefficient |
|---|---|
| Humor in AI | 0.8608 |
| Electronic Sheep | 0.8533 |

### B.14 PERFORMANCE WITH GPT4.1 AS THE EVALUATOR

We conducted additional experiments using GPT-4.1 (the second-strong evaluator in Table 2) to assess pass@k and statistical significance. Results corroborate our GPT-5 findings: HOMER consistently outperforms all baselines in most cases, with significant differences (p <0.05), as shown in Table 20 and Table 21.

**Agreement Evaluation.** We measure the agreement between GPT-4.1 scores and human ratings per caption. The results show that GPT-4.1 exhibits moderate positive alignment with humans with 0.5639 Pearson coefficient.

### B.15 ANALYSIS OF DATA LEAKAGE

**Different data sources and formats.** The joke database primarily comprises one-liner jokes sourced from Pun of the Day, TED Talks, Reddit (r/jokes), and short-joke websites, and explicitly excludes cartoon or image captions. In contrast, the test set consists of original, publicly submitted captions from the New Yorker Caption Contest.

**Different task usage.** The joke database is used exclusively for text-only humor tasks (e.g., humor detection, humor rating, and joke generation). The test set is reserved for a multimodal humor task, such as funny caption generation, ensuring clear separation of application contexts.

**Empirical negligible overlap.** We evaluate potential leakage by quantifying instance-level overlap between the test set and the joke database using the exact-match and normalized comparisons of caption answers. All metrics yielded zero overlap, indicating no evidence of leakage between the joke database and the testing data.

### B.16 ABLATION ON JOKE DATABASE

We have conducted experiments of ablation on the scale of our joke dataset, varying the proportion from 0% to 100%. At 0%, the model relies solely on the GTVH-guided structure. The results indicate that our model's performance increases steadily, further demonstrating the necessity of our joke database. Combined with the ablation studies in Table 4, the performance of the joke database alone, without our designed humor mechanism, shows a significant drop. These results validate both our novel GTVH-based framework and joke database.

## C THEORETICAL ANALYSIS OF $\mathcal{H}_{\text{rel}}$

We define the relevance-opposition score as

$$\mathcal{H}_{\text{rel}}(e_\tau^{(i)}, \varepsilon) = \text{TSS}(s_{e_\tau}, s_\varepsilon) + f(\text{TSS}(s_{e_\tau}, s_\varepsilon)) \cdot CO(s_{e_\tau}, s_\varepsilon), \tag{9}$$

Table 20: Pass@k results for the Humor in AI dataset across different subsets.

| Methods | pass@1 | pass@3 | pass@5 |
|---------|--------|--------|--------|
| **Top 10** | | | |
| CoT | 61.0 | 76.0 | 81.9 |
| Few-Shot | 67.8 | 76.5 | 82.4 |
| Self-Consistency | 69.6 | 90.1 | 92.7 |
| CLoT | 60.8 | 74.8 | 80.0 |
| HumorAI | 68.2 | 85.5 | 89.6 |
| LoL | 70.6 | 89.3 | 93.3 |
| Phunny | 16.8 | 32.5 | 42.0 |
| Our HOMER | 74.6 | 90.6 | 95.0 |
| **#200-109** | | | |
| CoT | 64.2 | 81.1 | 85.0 |
| Few-Shot | 71.6 | 87.9 | 90.0 |
| Self-Consistency | 69.0 | 86.0 | 88.9 |
| CLoT | 63.6 | 74.1 | 78.0 |
| HumorAI | 72.8 | 86.5 | 90.0 |
| LoL | 73.2 | 90.9 | 93.9 |
| Phunny | 17.8 | 31.1 | 38.0 |
| Our HOMER | 75.2 | 91.5 | 95.0 |
| **#1000-1009** | | | |
| CoT | 70.2 | 84.5 | 89.7 |
| Few-Shot | 73.8 | 89.2 | 90.8 |
| Self-Consistency | 73.8 | 88.2 | 91.0 |
| CLoT | 73.4 | 82.4 | 84.0 |
| HumorAI | 74.4 | 87.8 | 91.0 |
| LoL | 74.6 | 90.9 | 94.5 |
| Phunny | 25.4 | 38.4 | 45.0 |
| Our HOMER | 77.2 | 89.6 | 95.1 |

Table 21: Pairwise significance test results.

| Comparison Pair | p-value | Significant? ($\alpha = 0.05$) |
|-----------------|---------|--------------------------------|
| **Ours** vs. CoT | 0.0021 | Yes |
| **Ours** vs. Few-Shot | 0.0089 | Yes |
| **Ours** vs. Self-Consistency | 0.0091 | Yes |
| **Ours** vs. CLoT | 0.0030 | Yes |
| **Ours** vs. HumorAI | 0.0108 | Yes |
| **Ours** vs. LoL | 0.0142 | Yes |
| **Ours** vs. Phunny | 0.0000 | Yes |

where $\text{TSS}(\cdot, \cdot)$ measures semantic similarity and $CO(\cdot, \cdot)$ quantifies conceptual opposition, with $f(x) = x \exp(-x)$ serving as a similarity-gated modulation function. This formulation fulfills the following criteria:

(i) *Dominance by Semantic Similarity*: The term $\text{TSS}(s_{e_\tau}, s_\varepsilon)$ is the primary additive component, ensuring that the overall score increases monotonically with greater semantic similarity, regardless of the value of $CO$.

(ii) *Similarity-Gated, Bounded Bonus for Conceptual Opposition*: The $CO$ term is multiplied by $f(\text{TSS})$, which serves as an adaptive gate. Since $f(x) = x \exp(-x)$ achieves its maximum at $x = 1$ and decays to 0 as $x \to 0$ or $x \to \infty$, the contribution of conceptual opposition is substantial only for intermediate semantic similarity and is suppressed for both very low and very high TSS. Moreover, since $|f(x)|$ is maximized at $e^{-1}$, and $CO$ is assumed to be bounded (e.g., $|CO| \leq 1$), the bonus (or penalty) term is inherently bounded in magnitude.

(iii) *Principled Balance Between Competing Tendencies*: The function $f$ provides a smooth and principled balance between rewarding similarities and oppositions: it modulates the influence of

Table 22: Performance by database scale across different groups.

| Group | Scale of Joke DB | pass@1 (%) | pass@3 (%) | pass@5 (%) |
|---|---|---|---|---|
| **Top 10** | | | | |
| | 0% | 58.2 | 73.2 | 78.4 |
| | 25% | 63.4 | 82.0 | 87.1 |
| | 50% | 65.7 | 78.5 | 81.7 |
| | 75% | 66.1 | 80.3 | 87.6 |
| | **100%** | 66.4 | 83.7 | 89.2 |
| **#200-209** | | | | |
| | 0% | 60.3 | 75.1 | 80.8 |
| | 25% | 66.2 | 84.9 | 88.7 |
| | 50% | 69.0 | 86.3 | 91.4 |
| | 75% | 71.5 | 87.2 | 92.1 |
| | **100%** | 73.4 | 88.4 | 92.6 |
| **#1000-1009** | | | | |
| | 0% | 63.2 | 78.7 | 83.7 |
| | 25% | 69.4 | 81.8 | 85.7 |
| | 50% | 72.9 | 84.3 | 90.0 |
| | 75% | 74.6 | 86.2 | 93.8 |
| | **100%** | 76.3 | 90.5 | 94.2 |

opposition such that opposition is only beneficial when the two sentences are neither too similar nor completely unrelated, reflecting a nuanced interplay between similarity and opposition.

In summary, the score $\mathcal{H}_{\mathrm{rel}}$ is monotonic in TSS when $CO = 0$, bounded for all inputs, and expresses a principled, interpretable balance between semantic similarity and conceptual opposition.

## D   FAILURE ANALYSIS

HOMER struggles with purely formal or inherently non-humorous images, especially when script opposition is difficult to detect. These cases are challenging even for humans, and the lack of narrative content and clear humor cues results in captions with limited humor.

## E   STYLE CONTROL

Our modular architecture enables controlled stylistic conditioning through curated imagination trees and the design of instructions in prompts. For example, the imagination tree can retrieve relevant semantic ambiguities among jokes in the joke database. Then, the Generator can reinforce selected imaginative entities through explicit style directives guided by the designed instruction.

## F   RELATED WORKS

**Classical Computational Humor.** Computational humor, as a challenging branch of computational linguistics, employs computational methods to study humor (Binsted et al., 2006; Wang et al., 2025), mainly including humor recognition (Cattle & Ma, 2018; Liu et al., 2018a; Xie et al., 2021; Zhou et al., 2020; Zou & Lu, 2019; Liu et al., 2018b; Shang et al., 2021), humor explanation (Hessel et al., 2023; Patro et al., 2021; Amin & Burghardt, 2020), and humor generation (Amin & Burghardt, 2020; Weller et al., 2020; Yamane et al., 2021; Zargham et al., 2023) tasks. Classical computational humor research focused on rule-based, statistical approaches, and multimodal techniques for detecting and modeling humor across text, audio, and vision (Inácio et al., 2023; Amin & Burghardt, 2020; Yang et al., 2015; Chauhan et al., 2021; Christ et al., 2022; Hasan et al., 2019). Recently, incorporating LLM reasoning techniques (Zhou et al., 2024; 2025b) into computational humor has become an interesting and meaningful direction.

# G   PROMPT DESIGN

**Prompt design in conflicting script extractor.** We design the prompt to instruct the conflicting script extractor to analyze a script-opposition-central situation description as the contextual background of the funny caption and then systematically identify and analyze conflicting or incongruous elements in the image $I$. The prompt includes the definition of script opposition, GTVH theory and the description of the image. It to analyze and extract all relevant conflicting scripts that exist in the situation description $D$ and the image $I$. The detailed prompt can be found in Appendix Figure 12.

**Prompt design in hierarchical imaginator.** There are two views of the imagination. The local observation $O_{\text{loc}}$ is from the detailed situation description $D$, capturing fine-grained entities or unexpected features within the image (e.g., oversized cups, professional figures). The global observation $O_{\text{glob}}$ leverages the image $I$ to emphasize the obvious entities in the scene (e.g., cups, table). For each view, the detailed prompt can be found in Appendix Figure 13.

**Prompt design in caption generator.** The generation prompt is constructed by integrating the situation description $D$, the selected conflicting scripts $C$, the creative imagination path $P_i$ of selected humor target $t_i$, and the generation options $\Omega \in NS \times LA$, where $\Omega$ specifies the narrative strategy and linguistic style. Formally, the prompt can be represented as $\Phi(\mathcal{C}, D, P_i, \Omega)$, which is then fed into the LLM-based caption generator producing the final funny caption. The detailed prompt can be found in Appendix Figure 14.

# H   LLM USAGE CLAIM

In this paper, LLMs are utilized for polishing writing and are strictly confined to improving coherence, grammar, and style within textual content. No additional functionalities are incorporated.

Figure 12: Example Prompt for Conflict Script Extractor

**SYSTEM**
Based on the Script Opposition theory from the General Theory of Verbal Humor (GTVH), analyze the given cartoon description and identify two or more conflict scripts exists in the description. A script refers to a bundle of knowledge or expectations about a particular situation. Script opposition occurs when two conflicting scripts (i.e., scenarios, expectations, or frames) are brought into contrast within the description, creating a basis for potential humor. In your answer, list pairs of conflicting scripts (each as phrases or a short sentence) that are opposed or contrasted within the description. Just present the conflicting script pairs directly.

**USER**
**Description of the image:** Inside a grand stone-walled throne room, a king sits stiffly on an ornate throne, his face marked by a mix of weariness and unease...

**ASSISTANT**
1. Imminent threat vs. calm indifference. 2. Royal authority and power vs. vulnerability and danger...

---

Figure 13: Example Prompt for Hierarchical Imaginator

**(Global View)**

**SYSTEM**
Given conflict scripts and a cartoon, your task is to identify the main entities mentioned. For each identified entity, generate a logical chain of three relevant entities, each based directly on the previous one. Associations may include ingredients, containers, sources, related objects, or common companions. Output JSON: key is the entity, value is a list of three such imaginations.

**USER**
**Image:** Encoded cartoon image.
**Conflicting scripts:** 1. Imminent threat vs. calm indifference. 2. Royal authority and power vs. vulnerability and danger...

**ASSISTANT**
[king, crown, head]...

**(Local View)**

**SYSTEM**
Given conflict scripts and a cartoon description, your task is to identify the main entities mentioned. For each identified entity, generate a logical chain of three relevant entities, each based directly on the previous one. Associations may include ingredients, containers, sources, related objects, or common companions. Output JSON: key is the entity, value is a list of three such imaginations.

**USER**
**Description:** Inside a grand stone-walled throne room, a king sits stiffly on an ornate throne, his face marked by a mix of weariness and unease...
**Conflicting scripts:** 1. Imminent threat vs. calm indifference. 2. Royal authority and power vs. vulnerability and danger...

**ASSISTANT**
[throne, chair, table], [stone-walled throne room, chandelier, hanging]...

---

Figure 14: Example Prompt for Caption Generator

**SYSTEM**
Using the provided free-association chains, conflict scripts, and the cartoon description, generate a witty, funny and smart caption that spotlights the central incongruity and naturally combines key keywords in chains. Consider techniques such as narrative setups, linguistic styles and puns, but keep it short, concise and suitable as a cartoon tagline.

**USER**

**Description:** Inside a grand stone-walled throne room, a king sits stiffly on an ornate throne, his face marked by a mix of weariness and unease...

**Conflicting Scripts:** 1. Imminent threat vs. calm indifference. 2. Royal authority and power vs. vulnerability and danger...

**Free-associating chains of cartoon:**
[thread, chandelier, hanging], [...]

**ASSISTANT**
Caption: When you're hanging by a thread, but the intern's still on their trial period.

---

