# OpenReview forum: "On the Wings of Imagination: Conflicting Script-based Multi-role Framework for Humor Caption Generation"
_ICLR.cc/2026/Conference — ICLR 2026 Poster_

### Official Review · Reviewer_gAin · 2025-10-31

**Soundness:** 3
**Presentation:** 3
**Contribution:** 2
**Rating:** 4
**Confidence:** 3

**Summary:**

This paper proposes a new framework for generating humorous image captions, called HOMER. It consists of three LLM-based components. First, the Extractor derives a general description from the image and identifies objects that contradict normal expectations as the basis for humor generation. Next, the Imaginator expands the humorous space both deeply and broadly by constructing imagination trees using humor-relevant retrieval, then pruning them based on humor-relevance scores. These results are finally fed into the Caption Generator to produce the desired humorous caption.

**Strengths:**

- Compared to previous work, this paper breaks away from the inherent humor mechanisms of LLMs. Instead of enhancing the model's ability to generate humorous image captions through prompt engineering or task-specific fine-tuning, it innovatively designs three distinct modules to complete the humor caption generation progressively. I find the overall pipeline design to be reasonable and well-structured.
- The design of the Imaginator is quite novel, as it takes into account both the depth and breadth of imagination. Moreover, the pruning strategy based on humor relevance scores is also fairly reasonable.

**Weaknesses:**

- The construction of the imagination tree relies too heavily on the joke database. The construction process of the joke database is not mentioned in the paper.
- Maybe the range of data that this pipeline can handle is quite limited. (Details see Questions)

**Questions:**

- The quantitative experimental results are quite thorough, but the presentation of the generated humorous captions seems rather sparse. From the example shown in Figure 2, I feel that some of the generated captions don't quite match the original images—the humor may be too far-fetched. Could you provide more examples of the generated captions?
- I have some doubts about the generalizability of your pipeline. If there are no particularly abnormal elements in the image, is this method still applicable? And how is the humor basis determined in such cases?

---

> ### Author Response · Authors · 2025-11-22
> **Response to Reviewer gAin (1/3)**
>
> We are deeply grateful for the constructive and valuable feedback, and have significantly revised the paper based on your suggestions, with major changes highlighted in blue. We hope that the following clarifications are helpful for your concerns.
>
> ---
>
> ### **[W1] Detailed Construction Process of the Joke Dataset**.
>
> Thank you for this crucial question. We have revised **Lines 305-307 in Section 3 and Appendix B.1** to clarify the detailed construction of the joke dataset. In the following, we present our joke database in terms of database construction and resource summary.
>
> **1. Multi-stage Data Construction of our Retrieval Joke Dataset.**
> We construct a large-scale humor retrieval dataset of 335,570 jokes through a multi-stage curation process. This joke database aggregates and reorganizes 11 publicly available, humor-related datasets: Pun of a Day, Short Jokes, Reddit Jokes, rJoke, SemEval 2021 Task 7, TED Laughter, HumorNorm, CleanComedy, ShortJokes, CrowdTruth, and Dad Jokes. All source datasets are distributed under open licenses.
>
> As an initial filtering step, we discard non-funny jokes in these datasets based on their given humor ratings in the originating datasets. Subsequently, we perform data cleaning to remove noisy characters to ensure data quality. To further refine the corpus, we eliminate duplicate jokes as well as jokes that exhibit excessive textual similarity. After completing these operations, our finalized humor retrieval database comprises a total of 335,570 jokes.
>
> **2. Summary of Data in Joke Database**
>
> |Corpus|Data Format|Data Sources|Intended Use|
> |-|-|-|-|
> |Short Jokes|One-liner jokes|Various joke websites | Humor detection (text-only) |
> | Reddit Jokes | One-liner jokes | Reddit (r/Jokes) | Humor detection (text-only) |
> | Pun of the Day | One-liner jokes | Pun of the Day website | Humor detection/recognition (text-only) |
> | rJoke | One-liner jokes | Reddit (r/Jokes) | Humor analysis (text-only) |
> | SemEval 2021 Task 7 | One-liner jokes | SemEval 2021 Task 7 | Humor detection/rating (text-only) |
> | TED Laughter | Speech | TED Talks | Humor analysis (text-only) |
> | HumorNorm | Words | English lexical resources | Lexical humor norms (text-only) |
> | CleanComedy | One-liner jokes | Reddit, Twitter, other platforms | Humor analysis (text-only) |
> | CrowdTruth | One-liner jokes | Various joke websites | Humor analysis (text-only) |
> | Dad Jokes | One-liner jokes | Grin, Dad Joke It | Humor analysis (text-only) |
> | **Our Test Dataset** | Cartoons with captions | New Yorker Caption Contest | Funny captioning (multimodal) |

---

> ### Author Response · Authors · 2025-11-22
> **Response to Reviewer gAin (2/3)**
>
> ### **[W2&Q2] The Generalizability of Our HOMER Approach**
>
> Thank you for this constructive suggestion. We have enhanced the generalizability of our HOMER by clarifying the motivation, adding new experiments, and analyzing data distribution.
>
> Specifically, we discuss HOMER’s generalizability across diverse image domains and summarize its core pipeline in **Section 1 (lines 51-74)**. To further validate robustness, we conduct a _new experiment_ on the Meme benchmark spanning realistic, comic, synthetic, and cartoon images in **Section 3.3 (Table 5)**, which indicates HOMER significantly outperforms state-of-the-art baselines on this diverse set. We also analyze the test-set distribution of humor types, beyond abnormalities, indicating that HOMER can handle multiple humor mechanisms—such as unexpected logic, contextual incongruity, exaggeration, etc, in **Appendix B.1 (Table 8)**.
>
> **1. Evaluation on the Meme Dataset.** Our proposed **HOMER can handle not only cartoons but also realistic, synthetic, comic images, and so on.** We have conducted a new experiment to evaluate HOMER on a new public ImgFlip meme dataset, which contains a diverse range of scenes. We evaluate the generated meme captions against the ground truth meme captions. HOMER consistently outperforms all state-of-the-art competitors by approximately 5.4% on average, validating the effectiveness and powerful generalization ability of our HOMER across different visual domains.
>
> |**Model**|**pass@1**|**pass@3**|**pass@5**|
> |-|-|-|-|
> |CoT|74.33|86.12|95.83|
> |Few-Shot|66.67|81.67|91.67|
> |Self-consistency |70.00|90.83|91.67|
> |HumorousAI |75.00|80.00| 83.33|
> |LoL|71.67| 81.12| 97.54|
> | Phunny |21.67|31.67|33.33|
> | CLoT | 76.67| 88.33| 91.67|
> | **Our Method**| **83.33** | **96.67**|**98.86**|
>
> **2. Generalization Beyond Visual Abnormalities.** Moreover, our method generalizes far beyond images featuring obvious abnormalities. It effectively handles multiple humor mechanisms—such as _unexpected logic, contextual incongruity, exaggeration, role reversals, etc_.
>
> For the Humor in AI dataset,
> |Humor Basis| Occurrences | Percentage |
> |-|-|-|
> |Abnormalities|122|34%|
> |Unexpected Logic|70|19%|
> |Contextual Incongruity|67|18%|
> |Exaggeration| 51|14%|
> |Ambiguity |55|15%|
> |Total|365|100%|
>
> **3. For mundane or formal images,** our HOMER is applicable but not suitable. Those serious images, such as product photos, standard landscapes, and official logs, may fall outside the scope of multimodal humor generation because they lack the narrative depth or incongruity required for humor. Therefore, our method can generate a normal caption for such images.

---

> ### Author Response · Authors · 2025-11-22
> **Response to Reviewer gAin (3/3)**
>
> ### **[Q1] More Examples of Generated Captions**
>
> We appreciate this useful feedback to strengthen our work. We have added two new examples of generated captions in **Figure 6 (Section 3.4) and Figure 11 (Appendix D).** Moreover, we have provided enhanced explanations with details to illustrate why the generated captions are humorous in **Section 1 (Lines 80-87) and Section 3.4**. Specifically, we responded in detail as follows.
>
> **1. Detailed Clarification of Case 1.** In terms of references (e.g., person, tables, and chairs), the size of coffee cups is super large, which is not normal but enlarged. Thus,  the extractor identifies the key conflicting script, i.e.,  gigantic coffee cups vs. normal ones. The imaginator then builds an association chain from the oversized coffee cups to excessive milk (coffee → milk → cow), leveraging two typical pairs of <coffee, milk> and <milk, cow>. Let us take a look at the **ground truth humorous caption** of this case, "Could you please pass me a cow?", highlighting the coffee-cow association and validating its relevance. On the other hand, our GTVH-guided generator derives the caption, "HR says we can expense a cow now", playfully exaggerating workplace coffee consumption.
>
> This caption suggests that employees, drinking large quantities of coffee, humorously claim HR allows them to expense a whole cow for milk. The exaggeration aligns with both the image and relatable office culture, resulting in humor that feels natural and contextually relevant. Overall, our approach adds the caption with an HR reference, introducing irony and potentially greater humor.
>
>
> **2. Detailed Clarification of Case 2.** For **Case 2** in Figure 6, the core script opposition is _Communication from a typical colleague vs. Communication through a mini-dummy_ as the fundamental joke logic. The hierarchical imaginator derives "smaller man$\rightarrow$puppet$\rightarrow$working" imagination chain of humor target _Smaller man_. Finally, the GTVH‑guided generator generates the funny caption **Is it Bring Your Inner Critic to Work Day already?** that fuses idiom subversion with personification to resolve the visual incongruity in a fresh way.
>
>
> **3. More case studies in Figures 6 and 11.**
>
> We added a new "Case 3" in Figure 6 (Section 3.4). The conflicting script is the dangerous high-altitude work vs. the casual pizza delivery, forming the basis of the joke. The hierarchical imaginator derives two association chains: "great height → upper" and "pizza → crust". Finally, the GTVH‑guided generator generates the funny caption **Now that's what we call 'upper crust' delivery!**. This caption cleverly plays on the idiom “upper crust,” connecting the visual context of a pizza being delivered at a great height to the phrase’s meaning of high quality or elite status.
>
> We added a new example in Figure 11 (Appendix D). The core script opposition is professional behavior expected vs. unexpected horseplay, forming the basis of the joke. The hierarchical imaginator derives "cowgirl$\rightarrow$reins" imagination chain of humor target _cowgirl_. Finally, the GTVH‑guided generator generates the funny caption **Well, Janet, I admire your drive, but this isn’t exactly what we meant by 'taking the reins at work'.** that cleverly plays on a well-known workplace idiom, "taking the reins at work".

---

### Official Review · Reviewer_SKPc · 2025-10-31

**Soundness:** 2
**Presentation:** 2
**Contribution:** 3
**Rating:** 4
**Confidence:** 3

**Summary:**

This paper proposes a novel framework for humour generation based on the General Theory of Verbal Humour. This framework is composed of three modules: 1. conflicting scripts extraction to find the out-of-expectation target with both global and local views, 2) humour-relevance retrieval to construct the hierarchical imagination tree from their collected joke database, and 3) final script generation. The experimental results on two humor generation datasets validate the effectiveness of this framework.

**Strengths:**

1. Grounding on the GTVH humour theory, it generates humour from conflicting elements in a novel way.
2. And considering both the similarity and the conceptual opposition. Utilising an external joke database for "humour" element retrieval.

**Weaknesses:**

1. Gaps between the figure illustration and the paragraphs:
1.1. Figure 1 lists 1. visual understanding, 2. humour understanding, 3. humour imagination, and 4. stylistic expression, and they are marked with checks and crosses. This leads the reader to expect an assessment along these dimensions, while the study provides no qualitative or quantitative evaluation of them.

2. Confusing Delivery
2.1 abstract, line 13-17.
2.2 The last line on page 1.
2.3 line 100, "The goal of tackling this task is to assess ..."
2.4 "extract conflicting script", e.g., line 103. Where are the scripts to extract? It appears that the authors are referring to the "strange" elements in the image, which contradict the norm.
2.5 line 290-291, "To assess the reliable win rate ...  use Pass rate (pass@K) to measure the win rate "

3. The paper's theoretical grounding, fundamental humour theory, GTVH, should be introduced more. Only line 139 mentions it.


typos
line 82, ";" should be '.'
line 190, redundancy ")" and "}"

**Questions:**

1. Line 38, are there other types of images for humour generation, besides cartoons?
2. Figure 2, Where is the direct link between Coffee and Sugar (demonstrated in the Global view)?
How to decide the Coffee node rather than the Cup node? In this case, it should be Cups, as it could be coffee, tea, or other drinks. And the conflicting elements are the sizes of the cups. How to ensure that these terms make sense? Could you please explain the humour in case 1? As for me, case 2 is humorous.
3. Term-2, It seems this equation would find those common words with a high humour-frequency score. What about trying the TF-IDF idea?
4. Line 262, Are all the candidate targets utilised or integrated in the final caption? How are they connected with the final caption?
5. Table 1, 6044, why are there so many captions?
6. Table 2: How do you convert the global ranking into the pairwise ranking for the Human-in-AI dataset?
7. line 290-291, How to calculate the win rate? Which competes against which?
8. Human evaluation: How many data points are evaluated? They are not stated in the appendix, either.

---

> ### Author Response · Authors · 2025-11-22
> **Response to Reviewer SKPc (1/3)**
>
> We are deeply grateful for the constructive and valuable feedback, and have significantly revised the paper based on your suggestions, with major changes highlighted in blue. We hope that the following clarifications are helpful for your concerns.
>
> ---
>
> ### **[W1] Qualitative and Quantitative Evaluation based on Four Dimensions**
>
> We thank the reviewer for this insightful and interesting suggestion, and we have added new experiments to evaluate four key dimensions using quantitative metrics, which are also updated in **Appendix B.6** of our revised paper. Specifically, we evaluate 1) visual understanding, 2) humor understanding, 3) humor imagination, and 4) stylistic expression as follows.
>
> To assess _**visual and humor understanding, as well as stylistic expression**_, which require advanced linguistic and cognitive judgment, we employed GPT-5 as an expert evaluator. Specifically, for each image, GPT-5 ranks the humorous captions generated by eight methods based on the designed criteria of three dimensions, ranging from best (#1) to worst (#8). We report the average rank of each method (lower is better) to ensure a fair and consistent assessment. To guarantee the reliability of the LLM ranking, we repeat each ranking for 5 trials.
>
> In addition, for _**humor imagination**_, we employ two established metrics: n-gram diversity, which measures lexical variety, and NLI Diversity, which measures the percentage of non-entailing caption pairs judged by a natural language inference model. We use a widely adopted Roberta-large model as the judge for NLI diversity. Higher scores on these metrics reflect greater imaginative capacity, as more diverse captions span a wider range of humorous ideas and content, indicating stronger imaginative ability in the model’s outputs.
>
> The following tables of results on two datasets show that our HOMER method _consistently outperforms all state-of-the-art competitors across four dimensions._ It achieves the best (lowest) average rank and the highest diversity scores, validating the superiority of our novel HOMER model.
>
>
>
> For the Humor in AI
>
> | Model | Visual Understanding | Humor Understanding | Stylistic Expression |Humor Imagination|   |
> |-|-|-|-|-|-|
> | | Avg. Rank(↓)  | Avg. Rank(↓) | Avg. Rank (↓) | 3-gram (↑)| NLI Diversity(%)(↑)|
> | CoT | 5.5 | 4.8 | 5.6 |0.87   | 85.3  |
> | Few-Shot | 4.4 | 3.7 | 3.8 | 0.84| 81.6 |
> | Self-Consistency | 5.9 | 5.4 | 5.7 |0.88| 85.9|
> | HumorousAI | 4.1 | 4.9 | 4.5 |0.59   | 70.0  |
> | LoL | 4.9 | 4.6 | 4.4 |0.92| 89.3 |
> | Phunny | 5.9 | 5.6 | 5.4 |0.69| 81.9  |
> | CLoT | 4.5 | 3.8 | 4.3 |0.46| 51.5 |
> | **HOMER (Ours)** | **2.5** | **3.2** | **2.3** |**0.98**| **91.5**  |
>
> For the Electronic Sheep Dataset,
>
> | Model | Visual Understanding | Humor Understanding | Stylistic Expression | Humor Imagination||
> |-|-|-|-|-|-|
> | | Avg. Rank(↓)  | Avg. Rank(↓) | Avg. Rank (↓) | 3-gram (↑)| NLI Diversity(%)(↑)|
> | CoT | 3.8 | 4.3 | 4.2 |0.88   | 84.2  |
> | Few-Shot | 4.5 | 4.5 | 4.1 |0.89| 83.1 |
> | Self-Consistency | 4.2 | 4.5 | 4.8 |0.84 | 86.40|
> | HumorousAI | 6.9 | 6.7 | 6.1 |0.92   | 88.5  |
> | LoL | 4.5 | 4.6 | 5.7 |0.94| 91.9 |
> | Phunny | 6.8 | 6.8 | 6.9 | 0.89| 85.2  |
> | CLoT | 3.5 | 3.2 | 2.7 |0.83| 78.9 |
> | **HOMER (Ours)** | **1.8** | **1.4** | **1.5** |**0.98**| **92.2**  |
>
>
> ### **[W2&Q7] Clarification of Background, Humor Theory, and Metrics**
>
> Thank you for your valuable suggestions. We have revised and added more clarification as advised.
>
> _For 2.1_, we have revised **Lines 13-16 in the Abstract** to clarify the background of humor and multimodal humor generation.
>
> _For 2.2_, we have revised **Lines 51-74 and 80-87 (Section 1)** to clarify the example in Figure 1 (b), combined with GTVH humor theory.
>
> _For 2.3_, we have revised **Lines 141-142 (Section 2)** to fix the correct explanation of the goal of the funny caption generation task.
>
> _For 2.4_, we have revised **Lines 143-145 (Section 2)** to clarify the key idea of our HOMER. According to the GTVH humor theory, conflicting scripts refer to the relationship between two opposing or contrasting semantic frames, such as normal vs. abnormal or expected vs. unexpected scenarios. In practice, elements that are abnormal or unusual, exhibit unexpected logic, or present surprising incongruities can serve as potential conflicting scripts.
>
> _For 2.5_, we have revised and added more clarification in the **Lines 312-320 (Section 3), Equation 7, Appendix B.3** to explain the calculation of the pass@k metric. Briefly, Pass@k measures the probability that HOMER-generated humorous captions outperform human-written captions over multiple $k$ trials. Meanwhile, we against three human caption levels: top-10, ranks 200–209, and ranks 1,000–1,009, representing high-, mid-, and lower-ranked human captions.

---

> ### Author Response · Authors · 2025-11-22
> **Response to Reviewer SKPc (2/3)**
>
> ### **[W3] Clarification of Humor Theory**
>
> Thank you for this critical suggestion. We have revised and added **Lines 51-70 in Section 1, Lines 130-138 in Section 2, and Figures 6 (Section 3.4) and 11 (Appendix D)** to introduce and clarify the GTVH humor theory.
>
> Here, we give a brief introduction to GTVH humor theory. The General Theory of Verbal Humor (GTVH) models humor as the interaction of the following knowledge resources that jointly shape interpretation: Script Opposition, Situation, Target, Narrative Strategy, and Language. **The key idea of GTVH** producing humor, is to generate script opposition, which captures conflicts between semantic frames (scripts). It underlies humor by establishing expectations and then violating them, thereby enabling exaggeration or absurdity.
>
>
>
> ### **[Q1] Other Applicable Types of Images for HOMER**
>
> Thank you for this constructive suggestion. We have discussed the generalizability of HOMER in a rich set of image data types and summarized its principal pipeline (lines 51-74) in Section 1. We also conducted new generalization experiments on a diverse meme dataset, ImgFlip, in Section 3.3 and Table 5.
>
> Our proposed HOMER can also handle other image types, including realistic, synthetic, and comic images, and so on. We evaluate the generated meme captions against the ground truth meme captions. HOMER consistently outperforms all state-of-the-art competitors by approximately 5.4% on average, validating the effectiveness and powerful generalization ability of our HOMER across different visual domains.
>
> | **Model**|**pass@1**|**pass@3**|**pass@5**|
> |-|-|-|-|
> | CoT |74.33|86.12|95.83|
> | Few-Shot|66.67|81.67|91.67|
> | Self-consistency |70.00|90.83|91.67|
> | HumorousAI |75.00|80.00| 83.33|
> | LoL |71.67| 81.12| 97.54|
> | Phunny |21.67|31.67|33.33|
> | CLoT | 76.67| 88.33| 91.67|
> | **Our Method**  | **83.33** | **96.67**|**98.86**|
>
>
>
>
> ### **[Q2] Clarification of Figure 2**
>
> Thank you for this valuable question. We have revised the clarification of **Case 1 in Section 1 (Lines 80-87) and Section 3.4.** For the technical details, we revised **Lines 179-181, 195-199 in Section 2** to explain our method more clearly. We have also added and revised case studies in **Figure 6 and Figure 11, and Section 3.4. **
>
> Specifically, we first clarify your concern on the rationale of humor in Case 1 as follows.
>
> **The rationale of humor in Case 1 (Figure 6).** In terms of references (e.g., person, tables, and chairs), the size of coffee cups is super large, which is not normal but enlarged. Thus,  the extractor identifies the key conflicting script, i.e.,  gigantic coffee cups vs. normal ones. The imaginator then builds an association chain from the oversized coffee cups to excessive milk (coffee → milk → cow), leveraging two typical pairs of <coffee, milk> and <milk, cow>. Let us take a look at the **ground truth humorous caption** of this case, "Could you please pass me a cow?", highlighting the coffee-cow association and validating its relevance. On the other hand, our GTVH-guided generator derives the caption, "HR says we can expense a cow now", playfully exaggerating workplace coffee consumption. This caption suggests that employees, drinking large quantities of coffee, humorously claim HR allows them to expense a whole cow for milk. The exaggeration aligns with both the image and relatable office culture, resulting in humor that feels natural and contextually relevant. Overall, our approach adds the caption with an HR reference, introducing irony and potentially greater humor.
>
> Next, we clarify the technical details of your concerns in Figure 2.
>
> **Clarification of Imaginator in Figure 2.** Figure 2 illustrates a simplified example of the imagination process. Within this process, the imaginator integrates the local and global views of the LLM-driven associations by aligning identical entities and removing duplicates, as detailed in Lines 198-201.
>
> Therefore, (1) in the global view, "coffee and sugar" is incorporated into the richer local view of "coffee, milk, and sugar" by aligning overlapping entities. (2) The candidate targets also include "tea," "coffee cups," or other relevant items. We select "coffee" as an example in Figure 2. In HOMER, there is a pool of candidate targets. (3) The imaginator engages in creative imagination from multiple relevant anchors such as “coffee,” “tea,” or “cups,” which all make sense and align with the inherently creative and multifaceted nature of humor. Multiple targets represent diverse imaginative directions.
>
> The goal of our hierarchical imagination is to conduct creative and diverse imagination relevant to the image. To ensure diversity while preventing irrelevant or nonsensical imagination, we constrain the imagination using LLM-driven associations and a humor relevance score tied to the situation and conflicting scripts.
>
>
> Finally, we add **more case studies in Figure 6 （Section 3.4）and Figure 11 (Appendix D).**

---

> ### Author Response · Authors · 2025-11-22
> **Response to Reviewer SKPc (3/3)**
>
> ### **[Q3] Comparison of Our Humor-Frequency Score and TF-IDF**
>
> Thank you for the suggestion regarding TF-IDF. We have conducted and added new experiments in **Appendix B.5.**
>
> There is one main difference between the humor-relevance score and TF-IDF: TF-IDF **down-weights** globally common tokens, whereas our humor-relevance score **up-weights** tokens or humor concepts that are frequently used in relevant jokes to a specific situation and conflicting scripts. We aim to identify entities that are important and frequently used as reliable components for humor generation. Therefore, the goal of our humor-relevance score and tf-idf is opposite.
>
> We ablated the choice of TF-IDF and our humor-relevance score on the Humor in AI dataset. Our humor-frequency score outperforms a TF-IDF-based score shown below
>
> | Scoring Method |pass@1 | pass@3| pass@5 |
> |-|-|-|-|
> | **TF-IDF** | 63.00  |  79.83 | 86.33  |
> | **Humor-Frequency (Ours)**| 66.41  | 83.70  | 89.18  |
>
>
> ### **[Q4] Clarification of Detailed Method**
>
>
> We have revised **Lines 279-280 and Line 285 in Section 2** to explain the generator process of the final caption by selecting high-quality targets, where not all candidate targets are used in the final caption. A more detailed process can be found in **Appendix Algorithm 1 (Lines 33-38)**.
>
>
>
> For each caption generation, not all candidate targets are used. We design a structured, multi-step mechanism that selects and connects relevant targets as humor anchors in the generated caption:
>
> _**Step 1: Selection of Core Humorous Conflicting Script.**_ From the pool of identified conflicting scripts, we select one as the primary basis for humor. Since an image may contain multiple incongruities, our framework accommodates diverse creative possibilities by allowing each script to serve as a distinct humor basis.
>
> _**Step 2: Identification of Relevant Candidate Targets.**_ We select candidate targets that are most relevant to the chosen conflicting script. For instance, if the chosen script is normal coffee cups vs. oversized coffee cups, "coffee cups" would be chosen as the humor target rather than objects like “figured people.”
>
> _**Step 3: Search the Imaginative Tree of the Chosen Target and Select the Imaginative Chain.**_ Next, we traverse the imagination tree for the chosen target, enumerating all possible semantic chains from the root to the leaves. We then sample one imaginative chain from these options to ensure diversity and creativity in the resulting caption.
>
> _**Step 4: Humorous Caption Generation.**_ Finally, we integrate the selected conflicting script, imaginative chain of target, and relevant humor theory principles into a structured prompt. This prompt is provided to the LLM, which generates the final humorous caption.
>
> This mechanism ensures that each caption is contextually coherent, creatively diverse, and firmly anchored in humor theory.
>
>
> ### **[Q5&Q6] Clarification of Datasets**
>
> Thank you for these precise questions regarding our experimental setup. Here, we give a detailed explanation.
>
> 1. The number "6,044" of captions in Table 1 is the average number of human-written captions submitted for each cartoon in the New Yorker Caption Contest as collected in the dataset. For each cartoon, these ~6k captions are globally ranked by aggregated public votes.
>
>
> 2. Converting global ranking to pairwise evaluation. The global ranking orders ~6,000 captions per cartoon from funniest (#1) to least funny, based on public votes. For pairwise evaluation, we compare our model-generated caption to the human-written caption sampled from three quality ranking levels: top 10,  top 200–209, and top 1,000–1,009, representing high-, mid, and lower-ranked human captions. Each comparison is pairwise.
>
>
>
> ### **[Q8] Clarification of Human Evaluation**
>
> We have added detailed human evaluation settings in **Section 3.5, Lines 462-464, and revised Table 6, and Appendix B.8.**
>
> In our original submission, the human evaluation was conducted on a total of 3,360 data points, calculated as 40 cartoon images were evaluated by 12 human raters across 7 different caption generation methods (40 images × 12 raters × 7 methods). Thanks to reviewer hHh4's suggestions, we increased the human raters from 12 to 20. Consequently, our final human evaluation is now based on a significantly larger and more reliable set of 5,600 data points (40 images × 20 raters × 7 methods).
>
> | Method |Humor in AI |Electronic Sheep |
> | :--- | :---: | :---: |
> | CoT | 2.47 |2.20 |
> | Few-Shot | 2.96 | 2.56 |
> | Self-Consistency | 2.66 | 2.25 |
> | CLoT | 2.95 | 2.53 |
> | HumorousAI | 3.01 | 2.24 |
> | LoL | 3.16 | 2.40 |
> | **HOMER (Ours)** | 3.54 | 3.31 |
>
>
> ### **Typo** Thank you for your suggestion and careful review. We have corrected the identified typos and carefully proofread the manuscript to improve the presentation.

---

### Official Review · Reviewer_hHh4 · 2025-11-01

**Soundness:** 4
**Presentation:** 4
**Contribution:** 4
**Rating:** 8
**Confidence:** 4

**Summary:**

The paper proposes HOMER, a novel framework for generating humorous captions for cartoon images. The approach is grounded in the General Theory of Verbal Humor (GTVH) and employs three coordinated LLM-based roles: (1) a conflicting-script extractor that identifies incongruities in images, (2) a hierarchical imaginator that builds imagination trees through LLM associations and humor-relevance retrieval from a joke database, and (3) a caption generator that produces humor based on the extracted knowledge. The method is evaluated on New Yorker Cartoon datasets, showing improvements over baselines.

**Strengths:**

1. **Theoretically grounded approach**: Unlike prior work that relies on generic prompting or self-improvement, HOMER explicitly incorporates established humor theory (GTVH), providing better interpretability and control over the humor generation process.

2. **Novel hierarchical imagination mechanism**: The combination of deep-pattern LLM associations with broad-pattern humor-relevance retrieval is creative and well-motivated, expanding the creative space while maintaining relevance.

3. **Strong empirical results**: The paper demonstrates consistent improvements (average ~7% on pass@1) across multiple baselines and LLM backbones, with comprehensive evaluation including human studies showing statistical significance.

4. **Thorough experimental analysis**: The paper includes extensive ablations, hyperparameter analysis, harmful content detection, and human evaluation with appropriate inter-rater agreement metrics.

5. **Clear modular design**: The three-role framework provides good interpretability and allows for targeted improvements to individual components.

**Weaknesses:**

1. **Questionable humor quality in examples**: The showcased example "HR says we can expense a cow now" doesn't seem particularly funny, raising concerns about whether the quantitative improvements translate to genuinely humorous outputs. More compelling examples would strengthen the paper's claims.

2. **Limited scope**: The method is only evaluated on cartoon caption generation. It's unclear whether the approach generalizes to other humor formats (jokes, puns, situational comedy, etc.).

3. **Computational overhead**: The method requires multiple LLM calls (7 total per caption generation according to Table 7)

5. **Limited human evaluation**: Only 12 raters participated

6. **Complexity concerns**: The framework involves many components (WordNet for semantic relations, multiple scoring functions, GTVH knowledge resources) that may be difficult to reproduce or adapt.

**Questions:**

1. **Generalization**: How does HOMER perform on other humor generation tasks beyond cartoon captioning? Have you tested on joke completion, pun generation, or conversational humor?

2. **Example quality**: Could you provide more compelling examples that better demonstrate the humor quality? The current examples seem to lack the wit typically associated with New Yorker captions.

3. **Ablation on retrieval database**: What happens to performance as the joke database size varies? How sensitive is the method to the quality/style of jokes in the database?

4. **Failure analysis**: What types of images or situations does HOMER struggle with? Can you provide a systematic analysis of failure modes?

5. **Comparison fairness**: How many inference calls do the baseline methods use? Is the comparison fair given HOMER's computational overhead?

6. **Style control**: Can HOMER generate different styles of humor (e.g., dark humor, slapstick, wordplay) by modifying the retrieval database or prompts?

7. **Zero-shot performance**: How does HOMER perform without the joke retrieval component, relying only on the GTVH-guided structure?

---

> ### Author Response · Authors · 2025-11-22
> **Response to Reviewer hHh4 (1/3)**
>
> We are deeply grateful for the positive comments and insightful feedback, and have significantly revised the paper based on your suggestions, with major changes highlighted in blue. We hope that the following clarifications are helpful to address your concerns.
>
> ---
>
> ### **[W1&Q2] More Examples of Generated Captions**
>
> We appreciate this valuable feedback to strengthen our work. We have added two new examples of generated captions in **Figure 6 (Section 3.4) and Figure 11 (Appendix D).** Moreover, we have provided enhanced explanations with details to illustrate why the generated captions are humorous in **Section 1 (Lines 80-87) and Section 3.4.** Specifically, we responded in detail as follows.
>
> **1. Detailed Clarification of Case 1.** In terms of references (e.g., person, tables, and chairs), the size of coffee cups is super large, which is not normal but enlarged. Thus,  the extractor identifies the key conflicting script, i.e.,  gigantic coffee cups vs. normal ones. The imaginator then builds an association chain from the oversized coffee cups to excessive milk (coffee → milk → cow), leveraging two typical pairs of <coffee, milk> and <milk, cow>. Let us take a look at the **ground truth humorous caption** of this case, "Could you please pass me a cow?", highlighting the coffee-cow association and validating its relevance. On the other hand, our GTVH-guided generator derives the caption, "HR says we can expense a cow now", playfully exaggerating workplace coffee consumption.
>
> This caption suggests that employees, drinking large quantities of coffee, humorously claim HR allows them to expense a whole cow for milk. The exaggeration aligns with both the image and relatable office culture, resulting in humor that feels natural and contextually relevant. Overall, our approach adds the caption with an HR reference, introducing irony and potentially greater humor.
>
>
> **2. Detailed Clarification of Case 2.** For **Case 2** in Figure 6, the core script opposition is _Communication from a typical colleague vs. Communication through a mini-dummy_ as the fundamental joke logic. The hierarchical imaginator derives "smaller man$\rightarrow$puppet$\rightarrow$working" imagination chain of humor target _Smaller man_. Finally, the GTVH‑guided generator generates the funny caption **Is it Bring Your Inner Critic to Work Day already?** that fuses idiom subversion with personification to resolve the visual incongruity in a fresh way.
>
>
> **3. More case studies in Figures 6 and 11.**
>
> We added a new "Case 3" in Figure 6 (Section 3.4). The conflicting script is the dangerous high-altitude work vs. the casual pizza delivery, forming the basis of the joke. The hierarchical imaginator derives two association chains: "great height → upper" and "pizza → crust". Finally, the GTVH‑guided generator generates the funny caption **Now that's what we call 'upper crust' delivery!**. This caption cleverly plays on the idiom “upper crust,” connecting the visual context of a pizza being delivered at a great height to the phrase’s meaning of high quality or elite status.
>
> We added a new example in Figure 11 (Appendix D). The core script opposition is professional behavior expected vs. unexpected horseplay, forming the basis of the joke. The hierarchical imaginator derives "cowgirl$\rightarrow$reins" imagination chain of humor target _cowgirl_. Finally, the GTVH‑guided generator generates the funny caption **Well, Janet, I admire your drive, but this isn’t exactly what we meant by 'taking the reins at work'.** that cleverly plays on a well-known workplace idiom, "taking the reins at work".

---

> ### Author Response · Authors · 2025-11-22
> **Response to Reviewer hHh4 (2/3)**
>
> ### **[W2&Q1] The Generalizability of Our HOMER Approach**
>
>
> Thank you for this constructive suggestion. We have enhanced the generalizability of our HOMER by clarifying the motivation, adding new experiments, and analyzing data distribution.
>
> Specifically, we discuss HOMER’s generalizability across diverse image domains and summarize its core pipeline in **Section 1 (lines 51-74).** To further validate robustness, we conduct a new experiment on the Meme benchmark spanning realistic, comic, synthetic, and cartoon images in **Section 3.3 (Table 5)**, which indicates HOMER outperforms advanced baselines on this diverse set.
>
>
> **1. Evaluation on the Meme Dataset.** Our proposed **HOMER can handle not only cartoons but also realistic, synthetic, comic images, and so on.** We have conducted a new experiment to evaluate HOMER on a new public ImgFlip meme dataset, which contains a diverse range of scenes. We evaluate the generated meme captions against the ground truth meme captions. HOMER consistently outperforms all state-of-the-art competitors by approximately 5.4% on average, validating the effectiveness and powerful generalization ability of our HOMER across different visual domains.
>
> | **Model**|**pass@1**|**pass@3**|**pass@5**|
> |-|-|-|-|
> | CoT |74.33|86.12|95.83|
> | Few-Shot|66.67|81.67|91.67|
> | Self-consistency |70.00|90.83|91.67|
> | HumorousAI |75.00|80.00| 83.33|
> | LoL |71.67| 81.12| 97.54|
> | Phunny |21.67|31.67|33.33|
> | CLoT | 76.67| 88.33| 91.67|
> | **Our Method**  | **83.33** | **96.67**|**98.86**|
>
>
> **2. Clarification of our scope.** We revised our scope in **Lines 51-74.** Our study targets multimodal humor generation, which is fundamentally distinct from text-only tasks (e.g., joke completion or pun generation). The core challenge we address is mapping visual input to humorous text by understanding and extracting humor-relevant cues in images. Accordingly, text-only tasks fall outside the scope of our evaluation. While our implementation is multimodal, HOMER’s cognitive architecture, which includes Humor Understanding, Humor Imagination, and GTVH-guided Generation, can cross modalities and be instantiated for text-only settings by replacing the visual encoder with a textual context module. We consider this extension a promising area for future work.

---

> ### Author Response · Authors · 2025-11-22
> **Response to Reviewer hHh4 (3/3)**
>
> ### **[W3&Q5] Fair Comparison under Same LLM calls**
>
> Thank you for this crucial suggestion. We agree that a direct comparison of computational cost is essential. We have conducted and added a new matched-budget experiment (**updated in Appendix B.11**).
>
> For fairness, each baseline generates multiple independent humorous captions and selects the best one for each output. The number of attempts was set to match or exceed HOMER’s seven LLM calls. We show the pass@1 results evaluated by GPT-5 as follows. Despite equal or greater call budgets, baselines show modest gains and do not close the performance gap with HOMER, indicating that HOMER’s superiority is from its collaborative, structured framework rather than a higher computational budget.
>
> | Method  | LLM Calls per Output | pass@1 (%) |
> |-|-|-|
> | HumorousAI       | 9 (=3 calls × 3 repeats) | 62.7|
> | LoL     | 8 (=4 calls × 2 repeats) | 58.0|
> | Phunny  | 9 (=3 calls × 3 repeats) | 19.1|
> | CLoT    | 7 (=7 calls × 1 repeats) | 61.2|
> | **HOMER (Ours)** | **7**       | **66.4**     |
>
>
> **Principled use of multiple LLM calls.** In HOMER, multiple LLM calls are intrinsic to its structured, multi-agent design rather than a naive increase in computation. Each agent (extractor, imaginator, generator) performs a distinct, complementary function. This design yields an average improvement of approximately 7%, which we consider a reasonable cost–benefit trade-off.
>
>
>
> ### **[W4] More Human Evaluation**
>
> We appreciate the reviewer’s suggestion to strengthen the human evaluation. We have increased the number of raters from 12 to 20 and updated the new human evaluation results in **Section 3.5 Table 6 and Appendix B.8**.
>
> The updated results substantiate the superior performance of our method, enhancing the stability of the estimates.
>
> | Method |Humor in AI |Electronic Sheep |
> |-|-|-|
> | CoT | 2.47 |2.20 |
> | Few-Shot | 2.96 | 2.56 |
> | Self-Consistency | 2.66 | 2.25 |
> | CLoT | 2.95 | 2.53 |
> | HumorousAI | 3.01 | 2.24 |
> | LoL | 3.16 | 2.40 |
> | **HOMER (Ours)** | 3.54 | 3.31 |
>
>
> ### **[W5] Reproducibility**
>
> Thank you sincerely for the suggestions on reproducibility. We give the anonymized code link in the following comments.
>
>
>
>
>
> ### **[Q3&Q7] Ablation on Joke Database**
>
> Thank you for this constructive suggestion. We have conducted and added new experiments of ablation on the scale of our joke dataset in **Appendix B.15** as advised, varying the proportion from 0% to 100%. At 0%, the model relies solely on the GTVH-guided structure. The results indicate that our model’s performance increases steadily, further demonstrating the necessity of our joke database. Combined with the ablation studies in _Table 4_, the performance of the joke database alone, without our designed humor mechanism, shows a significant drop. **These results validate the effectiveness of both our novel GTVH-based framework and joke database.&&
>
> |  Group     | Scale of DB | pass@1(%)  | pass@3(%)  | pass@5(%)  |
> |-|-|-|-|-|
> | **Top 10**        |
> | | 0% | 58.2     | 73.2     | 78.4     |
> | | 25%| 63.4  |  82.0  |  87.1   |
> | | 50%| 65.7  | 78.5   |  81.7   |
> | | 75%| 66.1  | 80.3   |  87.6  |
> | | **100%**    |  66.4 | 83.7   |  89.2   |
> | **#200-209**      |
> | | 0% | 60.3   |  75.1  |  80.8   |
> | | 25%| 66.2  |  84.9  |  88.7   |
> | | 50%| 69.0   |  86.3  |  91.4   |
> | | 75%| 71.5   |  87.2  |  92.1   |
> | | **100%**    | 73.4  |  88.4  |  92.6   |
> | **#1000-1009**    |
> | | 0% |  63.2 |  78.7  |  83.7   |
> | | 25%| 69.4  |  81.8  |   85.7  |
> | | 50%|  72.9  |  84.3  |   90.0  |
> | | 75%| 74.6  |  86.2  |   93.8  |
> | | **100%**    | 76.3  |  90.5  | 94.2    |
>
>
>
> ### **[Q4] Failure analysis**
> Thank you for this insightful suggestion of failure analysis, and we have added the discussion of Failure analysis in Appendix E. HOMER struggles with purely formal or inherently non-humorous images, especially when script opposition is difficult to detect. These cases are challenging even for humans, and the lack of narrative content and clear humor cues results in captions with limited humor.
>
>
> ### **[Q6] Style control mechanism**
> We thank the reviewer for this interesting discussion, and we have added the discussion of style control in Appendix G. Our modular architecture enables controlled stylistic conditioning through curated imagination trees and the design of instructions in prompts. For example, the imagination tree can retrieve relevant semantic ambiguities among jokes in the joke database. Then, the Generator can reinforce selected imaginative entities through explicit style directives guided by the designed instruction.

---

> > ### Author Response · Authors · 2025-11-22
> > **Code Link for Reproducibility**
> >
> > Our demo code is available at the anonymized link (https://anonymous.4open.science/r/HOMER-demo-code) provided for review. We will publicly release the full data source after the review process.

---

### Official Review · Reviewer_VAjU · 2025-11-04

**Soundness:** 3
**Presentation:** 3
**Contribution:** 3
**Rating:** 6
**Confidence:** 3

**Summary:**

HOMER is a humor-theory–guided, three-role LLM for New Yorker-style cartoon captions that (1) extracts script oppositions, (2) builds retrieval-augmented “imagination” trees, and (3) generates captions; on two cartoon benchmarks it beats strong prompting baselines, with ablations showing each role matters.

**Strengths:**

1. Clear multi-role architecture grounded in humor theory; maps situation, script opposition, target, narrative strategy, and language to concrete modules
2. Consistent SOTA gains across datasets/groups and base models; effect sizes reported
3. Thorough ablations: module inclusion matrix (Tbl. 4) and score-term ablations (Fig. 3).
4. Toxicity audit across seven dimensions shows low scores

**Weaknesses:**

1. My biggest concern for methodology is for creative generation, the desireable generation can be diverse. Is there an way to enforce the diversity of the generated response and ensure all generated caption can beat the baseline. For a single type of humor generation, hacking the benchmark with certain prompt structure could still occur. In addition, the author should also measure the diversity for the generation to ensure that actual creative generation is produced.

2. Primary automatic metric uses GPT-5; while Table 2 benchmarks evaluators, no significance tests or cross-evaluator robustness are reported (e.g., bootstrap CIs on pass@K). Add statistical tests and report agreement between GPT-5 and humans per-caption.
3. Retrieval compliance & leakage: Retrieval set lists several joke corpora but lacks licensing and dedup/leakage controls vs. test captions. Report licenses, filtering, and overlap stats between retrieved text and ground-truth or generated captions.
4. Reproducibility gaps: Prompts, seeds, and full code release status are unclear.

**Questions:**

1. How sensitive are results to the choice of evaluator? Please report pass@K with a second strong evaluator and give 95% CIs; also correlate evaluator scores with human ratings per-caption

---

> ### Author Response · Authors · 2025-11-22
> **Response to Reviewer VAjU (1/3)**
>
> We are deeply grateful for the constructive and valuable feedback, and have significantly revised the paper based on your suggestions, with major changes highlighted in blue. We hope that the following clarifications are helpful for your concerns.
>
> ---
>
> ### **[W1] Diversity Evaluation**
>
> We agree that diversity is essential for creative generation, and we have conducted and added new diversity evaluation and analysis as advised (**updated in Appendix B.9**), demonstrating the effectiveness of our HOMER’s diversity strategy and the reliable evaluation on creative tasks. We explain it as follows.
>
>
> **1. Pass@k is a robust metric for evaluating creative generation.** All reported results (e.g., Table 2) use the Pass@k metric, which evaluates effectiveness across multiple, diverse generations. Particularly, pass@5 evaluates five distinct captions per image derived from each model, reflecting the reliable effectiveness of diverse captions as a whole. HOMER achieves a 4.77% gain over Humorous AI on pass@5, indicating that its five-caption sets outperform the baseline’s sets overall.
> _This provides a robust measure of creative reliability and inherently rewards diverse, high-quality generations._
>
>
> **2. New Evaluation of Diversity.** We further assess diversity using two established metrics: n-gram diversity, which measures lexical variety, and NLI Diversity, which measures the percentage of non-entailing caption pairs judged by a natural language inference model. We use a widely adopted Roberta-large model as the judge for NLI diversity. A higher diversity score indicates a more diverse set of captions. The results provide strong empirical evidence that HOMER can generate more diverse funny captions, consistently outperforming all baselines.
>
> For the Humor in AI Dataset
> | Model   | 1-gram (↑) | 2-gram (↑) | 3-gram (↑) | NLI Diversity (↑) |
> |-|-|-|-|-|
> | HumorousAI | 0.45  | 0.55  | 0.59   | 70.0%   |
> | LoL  | 0.64 | 0.83 | 0.87 | 89.3% |
> | Phunny    | 0.50  | 0.64  | 0.69 | 81.9%   |
> | CLoT  | 0.35 | 0.43 | 0.46 | 51.5% |
> | **Our Method**| **0.76**| **0.94**| **0.98**| **91.5%**|
>
> For the Electronic Sheep Dataset
> | Model   | 1-gram (↑) | 2-gram (↑) | 3-gram (↑) | NLI Diversity (↑) |
> |-|-|-|-|-|
> | HumorousAI | 0.65  | 0.87  | 0.92   | 88.5%   |
> | LoL  | 0.63 | 0.88 | 0.94 | 91.9% |
> | Phunny    | 0.64  | 0.84  | 0.89 | 85.2%   |
> | CLoT  | 0.58 | 0.78 | 0.83 | 78.9% |
> | **Our Method**| **0.72**| **0.94**| **0.98**| **92.2%**|
>
>
> **3. Designed Mechanism in HOMER for Enhancing Diversity.** The diversity results presented above substantiate the effectiveness of our HOMER’s diversity strategy, confirming that our method can generate more varied captions. The diversity mechanism includes _selecting different conflicting scripts and traversing the imagination tree_ to select different creative paths for each caption generation. This encourages the model to explore distinct humorous perspectives for the same image, reduces reliance on a single prompt pattern, and yields more diverse outputs.

---

> ### Author Response · Authors · 2025-11-22
> **Response to Reviewer VAjU (2/3)**
>
> ### **[W2&Q1] Significance Test and Agreement Evaluation between Evaluators and Human**
>
>
> Thank you for this valuable suggestion. We have added significance test and agreement evaluation in **Appendix B.12 and B.13.**
>
> **1. Significance test and Agreement Evaluation with GPT-5 as the Evaluator**
>
> Following your suggestions, we assess statistical significance with a two-sided Wilcoxon signed-rank test on Pass@k scores and quantify agreement between GPT-5 and human judgments via correlation analysis. The results show that our method significantly outperforms all baselines (p < 0.05) and that GPT-5’s evaluations are positively correlated with human ratings.
>
> **New significance test.** Following your advice, we use a 95% confidence interval and set the significance level α=0.05. A p-value < 0.05 indicates that the difference is statistically significant.
>
> For the Humor in AI dataset
> | Comparison Pair |  p-value     | Significant? (α=0.05) |
> |-|-|-|
> | Ours vs. CoT | 0.0000 | Yes    |
> | Ours vs. Few-Shot | 0.0027 |   Yes   |
> | Ours vs. Self-Consistency | 0.0031  | Yes |
> | Ours vs. CLoT | 0.0153 | Yes   |
> | Ours vs. HumorAI | 0.0036 |   Yes   |
> | Ours vs. LoL | 0.0001 | Yes |
> | Ours vs. Phunny | 0.0000  | Yes  |
>
> For the Electronic Sheep dataset
> | Comparison Pair |  p-value     | Significant? (α=0.05) |
> |-|-|-|
> | Ours vs. CoT | 0.0000 | Yes    |
> | Ours vs. Few-Shot | 0.0120 |   Yes   |
> | Ours vs. Self-Consistency | 0.0001  | Yes |
> | Ours vs. CLoT | 0.0011 | Yes   |
> | Ours vs. HumorAI | 0.0001 |   Yes   |
> | Ours vs. LoL | 0.0004 | Yes |
> | Ours vs. Phunny | 0.0000  | Yes  |
>
>
> **New agreement evaluation.** We measure the agreement between GPT-5 scores and human ratings per caption via the Pearson correlation coefficient. The results show that GPT-5 and humans are positively strongly correlated.
>
> | Evaluator Pair | Pearson coefficient |
> |-|-|
> | Humor in AI | 0.8608 |
> | Electronic Sheep| 0.8533|
>
>
>
> **2. Significance test and Agreement Evaluation with Second Strong Evaluator**
>
> We conduct additional experiments using GPT-4.1 (the second-strong evaluator in Table 2) to assess pass@k and statistical significance. Results corroborate our GPT-5 findings: HOMER consistently outperforms all baselines in most cases, with significant differences (p < 0.05).
>
> **Performance with GPT 4.1 as the Evaluator**
>
> For the Humor in AI dataset
> | Methods | pass@1|pass@3|pass@5|
> |-|-|-|-|
> |Top 10| | | |
> |  CoT | 61.0 | 76.0 | 81.9 |
> |  Few-Shot |  67.8 | 76.5| 82.4|
> |  Self-Consistency | 69.6  | 90.1 | 92.7|
> |  CLoT | 60.8  | 74.8| 80.0|
> |  HumorAI | 68.2 | 85.5| 89.6 |
> |  LoL | 70.6| 89.3 | 93.3|
> |  Phunny | 16.8 | 32.5 | 42.0|
> |  Our HOMER| 74.6 | 90.6 | 95.0|
> |#200-109| | | |
> |  CoT | 64.2 | 81.1 | 85.0 |
> |  Few-Shot | 71.6 | 87.9 | 90.0 |
> |  Self-Consistency | 69.0 | 86.0 | 88.9 |
> |  CLoT | 63.6 | 74.1 | 78.0 |
> |  HumorAI | 72.8 | 86.5 | 90.0 |
> |  LoL | 73.2 | 90.9 | 93.9 |
> |  Phunny |17.8 |31.1 |38.0 |
> |  Our HOMER |75.2 | 91.5 | 95.0 |
> |#1000-1009| | | |
> |  CoT | 70.2 | 84.5 | 89.7|
> |  Few-Shot |73.8  |89.2 | 90.8 |
> |  Self-Consistency | 73.8 | 88.2 | 91.0 |
> |  CLoT | 73.4  | 82.4 | 84.0 |
> |  HumorAI | 74.4 | 87.8 | 91.0|
> |  LoL | 74.6 | 90.9| 94.5|
> |  Phunny |25.4 |38.4 | 45.0|
> |  Our HOMER| 77.2| 89.6| 95.1 |
>
> **Significance Test Results**
>
> | Comparison Pair |  p-value     | Significant? (α=0.05) |
> |-|-|-|
> | Ours vs. CoT | 0.0021 | Yes    |
> | Ours vs. Few-Shot | 0.0089 |   Yes   |
> | Ours vs. Self-Consistency | 0.0091  | Yes |
> | Ours vs. CLoT | 0.0030 | Yes   |
> | Ours vs. HumorAI | 0.0108 |   Yes   |
> | Ours vs. LoL | 0.0142 | Yes |
> | Ours vs. Phunny | 0.0000  | Yes  |
>
>
> **New agreement evaluation**
>
> We measure the agreement between GPT-4.1 scores and human ratings per caption. The results show that GPT-4.1 exhibits moderate positive alignment with humans.
>
> | Evaluator Pair | Pearson coefficient |
> |-|-|
> |Humor in AI|0.5639|

---

> ### Author Response · Authors · 2025-11-22
> **Response to Reviewer VAjU (3/3)**
>
> ### **[W3] No Evidence of Leakage between Joke Dataset and Tested Dataset**
>
> We thank the reviewer for highlighting this key consideration, which is crucial for fair comparison. We have added new experiments of leakage and data analysis in **Appendix B.1 and B.14**, and found no evidence of data leakage between our joke database and the test set.
>
>
> **1. Different data sources and formats.** The joke database primarily comprises one‑liner jokes sourced from Pun of the Day, TED Talks, Reddit (r/jokes), and short‑joke websites, and explicitly excludes cartoon or image captions. In contrast, the test set consists of original, publicly submitted captions from the New Yorker Caption Contest.
>
> **2. Different task usage.** The joke database is used exclusively for text‑only humor tasks (e.g., humor detection, humor rating, and joke generation). The test set is reserved for a multimodal humor task, such as funny caption generation, ensuring clear separation of application contexts.
>
> | Corpus | Data Format | Data Source(s) | Intended Use |
> |-|-|-|-|
> | Short Jokes | One-liner jokes | Various joke websites | Humor detection (text-only) |
> | Reddit Jokes | One-liner jokes | Reddit (r/Jokes) | Humor detection (text-only) |
> | Pun of the Day | One-liner jokes | Pun of the Day website | Humor detection/recognition (text-only) |
> | rJoke | One-liner jokes | Reddit (r/Jokes) | Humor analysis (text-only) |
> | SemEval 2021 Task 7 | One-liner jokes | SemEval 2021 Task 7 | Humor detection/rating (text-only) |
> | TED Laughter | Speech | TED Talks | Humor analysis (text-only) |
> | HumorNorm | Words | English lexical resources | Lexical humor norms (text-only) |
> | CleanComedy | One-liner jokes | Reddit, Twitter, other platforms | Humor analysis (text-only) |
> | CrowdTruth | One-liner jokes | Various joke websites | Humor analysis (text-only) |
> | Dad Jokes | One-liner jokes | Grin, Dad Joke It | Humor analysis (text-only) |
> | Our Test Dataset | Cartoons paired with captions | New Yorker Caption Contest | Funny captioning (multimodal) |
>
> **3. Empirical negligible overlap.**
> We evaluate potential leakage by quantifying instance-level overlap between the test set and the joke database using the exact-match and normalized comparisons of caption answers. All metrics yielded zero overlap, indicating no evidence of leakage between the joke database and the testing data.
>
>
> **4. License and curation policy:** All datasets used are publicly available and we follow a strict curation protocol to prevent cross‑modal leakage: (i) we exclude any datasets that are multimodal or have been used for multimodal applications; (ii) the corpus is restricted to text‑only humor, focusing on short, one‑liner jokes with simple structure; and (iii) we will remove items that are near‑duplicates or overly similar to content in our multimodal captioning evaluation, minimizing any risk of overlap.
>
>
> ### **[W4] Reproducibility**
> Thank you sincerely for the suggestions on reproducibility. We give the anonymized code link in the following comments.

---

> > ### Author Response · Authors · 2025-11-22
> > **Code Link for Reproducibility**
> >
> > Our demo code is available at the anonymized link (https://anonymous.4open.science/r/HOMER-demo-code) provided for review. We will publicly release the full data source after the review process.

---

### Author Response · Authors · 2025-12-03
**Final Remarks by Authors**

Dear Chairs,

We sincerely thank all reviewers for their constructive and valuable feedback on our submission, which acknowledges the main contributions of our work, *HOMER*, as summarized below:

- **Innovative multi-role framework** (Reviewers VAjU, hHh4, SKPc, gAin)
- **Novel hierarchical imagination mechanism** (Reviewers hHh4, SKPc, gAin)
- **Solid theoretical foundation** (Reviewers VAjU, hHh4, SKPc)
- **Extensive experimental evaluation and ablations** (Reviewers VAjU, hHh4, gAin)
- **Strong experimental and empirical results** (Reviewers VAjU, hHh4, SKPc, gAin)


We deeply appreciate the reviewers’ time and dedication. Based on their comments, we identify two primary concerns: **(1) explanation of case studies**, and **(2) generalization across different types of images**. Moreover, we clarify **more details**.

During our rebuttal comments, we addressed both with additional experiments and analyses, as outlined below.

- **Case explanation:** We conducted detailed case studies to analyze the inherent humor logic in our generated funny captions building on the derived conflicting scripts and humor imagination guided by humor theory (in response to Reviewer hHh4 [W1, Q2], Reviewer SKPc [Q2], and Reviewer gAin [Q1]), validating HOMER’s strong performance of funny caption generation and representing the **controllable and explainable** mechanism of HOMER. Particularly, for the mentioned Case 1/Figure 2, our generated caption is **consistent** with the ground truth humorous caption of the coffee-cow imagination. In Figure 6 (Section 3.4) and Figure 11 (Appendix D), we add two new examples of generated captions and enhance explanations with details to illustrate why the generated   captions are humorous in Section 1 (Lines 80-87) and Section 3.4.


- **Generalization:** In Section 1 (lines 51-77), we add discussions across diverse image domains and summarize its core pipeline. Furthermore, in Section 3.3 (Table 5), we add new experiments to evaluate HOMER on diverse images, including *realistic, comic, synthetic, cartoon images and so on* (in response to Reviewer hHh4 [W2, Q1], Reviewer SKPc [Q1], and Reviewer gAin [W2, Q2]. Across a diverse range of scenes, **HOMER consistently shows robust generalization and maintains state-of-the-art performance**.


**Clarification of details:**


- We clarify the details of our joke database in Section 3 and Appendices B.1&B.14 and add ablations in Appendix B.5 (in response to Reviewer gAin [W1], Reviewer hHh4 [Q3, Q7], Reviewer VAjU [W3])

- We clarify the details of our HOMER, humor theory, and human evaluation in Abstract, Sections 2&3, and Appendix B.3&B.8 and release our codes (in response to Review SKPc [W2, W3, Q4, Q7-8], Reviewer hHh4 [W4, W5], Reviewer VAjU [W4])

- We add further experiments according humor dimensions and diversity in Appendices B.6&B.9 (in response to Reviewer SKPc [W1], Reviewer VAjU [W1])

- We add more ablations and discussions in Appendices B.5&B.11&B.12&B.13&B.15&E&G (in response to Reviewer SKPc [Q3], Reviewer VAjU [W2, Q1], Reviewer hHh4 [W3, Q4-6])



These additional results have been carefully integrated into the revised paper, along with deeper explanations. We respectfully please Chairs consider these clarifications.

We once again extend our sincere gratitude to the Chairs and all reviewers for their time and dedication.

Best regards,

Authors of Submission 5023

---

### Meta-Review · Area_Chair_o19U · 2025-12-30

**Summary:**

This paper presents a novel framework for generating humorous captions for cartoon images, grounded in the General Theory of Verbal Humor (GTVH). The proposed approach decomposes humor generation into three LLM)–based roles: (1) a conflicting-script extractor, (2) a hierarchical imaginator that constructs imagination trees and (3) a caption generator.

All reviewers unanimously acknowledge the novelty of the multi-role framework. In particular, the hierarchical imagination mechanism is regarded as a well-designed and original contribution, and the integration of GTVH provides a theoretically grounded foundation.

The experimental evaluation is also well acknowledged as extensive and carefully conducted.

Two primary concerns were raised during the review process: (1) the clarity and depth of explanation in the qualitative case studies, and (2) the generalization of the approach across different types of images. In the rebuttal and revised manuscript, the authors have clarified the case study analyses, adding additional experiments along humor dimensions and diversity, and providing further ablation studies and discussions in the appendices. These additions should address the reviewers’ concerns.

Overall, this paper offers a well-motivated, theoretically grounded, and technically innovative solution to humorous caption generation, supported by strong empirical evidence. The revisions adequately resolve the raised issues, and the contribution is clear and significant. Therefore, it is recommended for acceptance.

**Reviewer Concerns:**

VAjU Diversity Evaluation  clarified with  added new diversity evaluation and analysis

no significance tests or cross-evaluator robustness are reported:  Rebuttal adds them.

hHh4 the clarity and depth of explanation. More examples are provided.

SKPc   Asking some details. Addressed with rebuttal.

gAin  the generalization of the approach across different types of images. Solved with new experiments on diverse images,

**Reviewer Scores:**

VAjU may remain the same or increase from 6

hHh4 No change.

SKPc  increase from 4 to 6

gAin  increase from 4 to 6

---

### Decision · Program_Chairs · 2026-01-26

Accept (Poster)